# Out-of-Variable Generalization for Discriminative Models

**Siyuan Guo** [*, †, ‡]  **Jonas Wildberger** [†]  **Bernhard Schölkopf** [†]

## Abstract

The ability of an agent to do well in new environments is a critical aspect of intelligence. In machine learning, this ability is known as *strong* or *out-of-distribution* generalization. However, merely considering differences in distributions is inadequate for fully capturing differences between learning environments. In the present paper, we investigate *out-of-variable* generalization, which pertains to an agent's generalization capabilities concerning environments with variables that were never jointly observed before. This skill closely reflects the process of animate learning: we, too, explore Nature by probing, observing, and measuring proper *subsets* of variables at any given time. Mathematically, *oov* generalization requires the efficient re-use of past marginal information, i.e., information over subsets of previously observed variables. We study this problem, focusing on prediction tasks across environments that contain overlapping, yet distinct, sets of causes. We show that after fitting a classifier, the residual distribution in one environment reveals the partial derivative of the true generating function with respect to the unobserved causal parent in that environment. We leverage this information and propose a method that exhibits non-trivial out-of-variable generalization performance when facing an overlapping, yet distinct, set of causal predictors. Code: https://github.com/syguo96/Out-of-Variable-Generalization

## 1 Introduction

Much of modern machine learning can be viewed as large-scale pattern recognition on suitably collected *independent and identically distributed (i.i.d.)* data. Its success builds on generalizing from one observation to the next, sampled from the same distribution. Animate intelligence differs from this in its ability to generalize from one *problem* to another. The machine learning community studies the latter under the term *out-of-distribution* (OOD) generalization (Shen et al., 2021; Parascandolo et al., 2021; Ahuja et al., 2021; Krueger et al., 2021; Zhang et al., 2021b;a; Schölkopf, 2022), where training and test data differ in their distributions. However, differences in distributions do not fully capture differences in environments. In the present work, we investigate generalization across environments where different sets of variables are observed, referring to the problem as **out-of-variable (OOV)** generalization. While in practice we would expect that many real-world situations exhibit both aspects of OOD and OOV, we note that the OOV problem can occur even if there is no shift in the underlying distribution. In the present paper, we will focus on this setting.

OOV generalization aims to transfer knowledge learnt from a set of source environments to a target environment that contains variables never jointly present in any of the sources, or even not present at all. OOV is a ubiquitous problem in inference. Scientific discovery synthesizes information and generalizes both out-of-distribution and out-of-variable (Seneviratne et al., 2018; Hey et al., 2009). Medicine is a field where machine learning is thought to have great potential since it can learn from millions of patients while a doctor may only see a few thousand during their lifetime. However, we face strong limitations in guaranteeing dataset consistency. To begin with, patients have unique circumstances, and some diseases/symptoms are rare. More generally, medical datasets come with different variable sets —- diagnostic measurements greatly vary across patients (serum, various forms of imaging, genomics, proteomics, immunoassays, etc.). A good doctor, however, will be able to generalize across patients even if the measured variables are not identical. In practice, data scientists

---

[*]Correspondence to: siyuan.guo@tuebingen.mpg.de

[†]University of Cambridge, United Kingdom

[‡]Max Planck Institute for Intelligent Systems, Tübingen, Germany

end up using imputation or simply ignoring rarely-measured features. To realize the potential of AI in medicine, we need to understand the OOV problem.

To provide context, we briefly discuss pertinent research threads.

**Missing data** Rubin (1976) refers to when covariates are missing for individual data points. Most approaches (Donner, 1982; Kim and Curry, 1977) either omit data that contain missing values or perform imputation. Our problem differs in that some variables are missing in entire environments.

**Transfer learning** studies how to re-use previous knowledge for future tasks. Recent work focused on transferring re-usable features (Long et al., 2015; Oquab et al., 2014; Tzeng et al., 2015) or model parameters (Dodge et al., 2020; Sermanet et al., 2013; Hoffman et al., 2014; Cortes et al., 2019; Wenzel et al., 2022) of discriminative models. Deep learning based approaches (Meyerson and Miikkulainen, 2020; Reed et al., 2022) embed variable relationships as proximity in the latent spaces. Our work presents a theoretical study on OOV generalization showing that without additional assumptions, the discriminative OOV problem is not solvable: marginal consistency between source and target discriminative models does not uniquely determine a solution.

**Marginal problems** in the statistical (Vorob'ev, 1962; Sklar, 1996) and causal literature (Mejia et al., 2021; Gresele et al., 2022; Janzing, 2018; Janzing and Schölkopf, 2010; Evans and Didelez, 2021; Robins, 1999), on the other hand, study how to merge marginal information from different sources. Concrete methods may involve searching for a joint distribution that is consistent with marginal observations. The elegant work of Mejia et al. (2021) uses the maximum entropy principle to infer joint distributions compatible with observed marginal datasets; Gresele et al. (2022) study the existence of consistent causal models; Janzing (2018) aims to learn useful causal models that can predict properties of previously unobserved variable sets. Note that inferring the joint distribution for prediction tasks may be inefficient, in line with Vapnik's principle (Vapnik, 1999): given some task, one should avoid solving a more general problem as an intermediate step. Our work takes a different approach, showing that learning from a residual error distribution is sufficient to achieve identifiability in nontrivial discriminative OOV scenarios.

**Causality** has been argued to be related to the issue of generalization across domains (Zhang et al., 2015; Schölkopf et al., 2011; Arjovsky et al., 2019; Pearl and Bareinboim, 2022). Distribution shifts between domains can be modelled as sparse causal mechanism shifts (Bengio et al., 2019; Schölkopf, 2022; Perry et al., 2022) , and the correct causal structure may help efficient modular adaptation (Parascandolo et al., 2018; Goyal et al., 2020). Other work considers domain differences as shifts in spurious correlations or aims to learn invariant causal information, robust across environments (Schölkopf et al., 2011; Peters et al., 2016; Rojas-Carulla et al., 2018; Heinze-Deml et al., 2018; Arjovsky et al., 2019; Jiang and Veitch, 2022; Krueger et al., 2021; Parascandolo et al., 2021; Ahuja et al., 2021; Lu et al., 2021; Heinze-Deml et al., 2018; Pfister et al., 2019; Rojas-Carulla et al., 2018). Causality approaches also include the transfer of causal effects across different experimental conditions (Pearl and Bareinboim, 2022; Bareinboim and Pearl, 2013; 2016; Degtiar and Rose, 2023). In the present paper, we highlight another connection between causality and generalization, where domain differences are due to distinct sets of variables contained within them, and causal assumptions allow us to generalize even to variables that we have never seen during training.

*Out-of-variable* generalization studies the efficient re-use of marginal observations. We do not *solve* this problem, neither do we present a robust algorithm for real-world settings. We do present a proof-of-concept, proposing a setting and a predictor provably capable of leveraging additional information beyond what is typically used by discriminative models. We do so without the need for inferring joint distributions. Our main contributions are:

- We contextualize (§ 1) and study OOV generalization (§ 2).

- We investigate challenges for common approaches (e.g., transferring reusable features or model parameters) to solve discriminative OOV problems (§ 3.2). We show (Theorem 1) marginal consistency condition alone does not permit identification of the target predictive function for common OOV scenarios.

- We study the identification problem in OOV scenarios when source and target covariates have dependent (§ 3.3.1) and independent (§ 3.3.2) structures. We find that the moments of the error distribution in the source domain reveal the partial derivative of the true generating function with respect to the unobserved causal parents (§ 3.3.4).

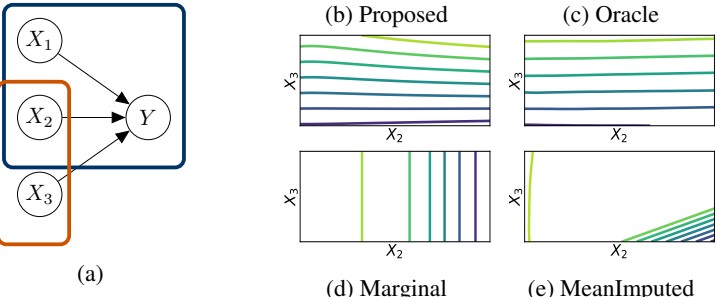

Figure 1: Example of an OOV scenario: (a) the blue box includes observed variables in the source domain, and the orange box those in the target domain. A directed edge represents a causal relationship. With $Y$ not observed in the target domain, the goal is to predict $Y$ in the target domain using the source domain. (b)-(e) shows an example of contour lines of various methods' prediction on $\mathbb{E}[Y \mid X_2, X_3]$. Our proposed predictor (b) results in a close match with the true expectation (c), an oracle solution trained as if we have sufficient data and observe all variables of interest. In contrast, marginal and mean imputed predictors (d, e) deviate far from the true expectation (for details, cf. § 4).

- We then propose an OOV predictor and evaluate its performance experimentally (§ 4), showing that our approach achieves a non-trivial degree of OOV transfer.

Fig. 1 provides a toy example of our problem. At first glance, it would seem all but impossible to have any transfer from the source environment (blue box) to the target environment (orange box) about information on an unobserved variable in the source. The goal of the present paper is to show that under certain causal and functional assumptions, there is a previously overlooked source of information in this OOV setting, making it possible after all.

## 2 OUT-OF-VARIABLE GENERALIZATION

Denote by $X$ a random variable with values $x$, $P$ a probability distribution with density $p$. Consider an acyclic structural causal model (SCM) $\mathcal{M}$ consisting of a collection of random variables and structural assignments (Pearl, 2009)

$$X_i := f_i(\mathbf{PA}_i, U_i), \ i = 1, \ldots, n, \tag{1}$$

where $\mathbf{PA}_i$ are the parents or direct causes of $X_i$ and $U_i$ are jointly independent noise variables. Given an SCM $\mathcal{M}$, one can define its corresponding directed acyclic graph (DAG) where the incoming edges for each node are given by its parent set. A joint distribution generated from some SCM $\mathcal{M}$ with DAG $\mathcal{G}$ allows the **Markov factorization**

$$p(x_1, \ldots, x_n) = \prod_{i=1}^{n} p(x_i \mid \mathbf{pa}_i^{\mathcal{G}}) \tag{2}$$

where $\mathbf{pa}_i^{\mathcal{G}}$ are parents of $X_i$ in $\mathcal{G}$. The factors ("mechanisms") in (2) are postulated to be independent:

**Principle 1** (Independent Causal Mechanisms (ICM) (Peters et al., 2017)). *A change in one mechanism $p(x_i \mid \mathbf{pa}_i^{\mathcal{G}})$ does not inform (Guo et al., 2022; Janzing and Schölkopf, 2010) or influence (Schölkopf et al., 2011) any of the other mechanisms $p(x_j \mid \mathbf{pa}_j^{\mathcal{G}})(i \neq j)$.*

Before defining *OOV* generalization, we begin by motivating the problem. Probabilistic representations (such as the Markov factorization Eq. 2) have been argued to offer advantages for probabilistic inference (Koller and Friedman, 2010) and interpretability. We highlight that in addition, *the Markov factorization frees us from the need of observing all variables of interest at the same time:*

**Observation** (Estimating the joint via causal modules). *Suppose that we have knowledge of the causal DAG and would like to estimate the joint density $p$. Provided that for each variable $X_i$, we observe an environment containing $X_i$ and its causal parents, we can recover the joint density by multiplying (according to Eq. 2) the conditionals $p(x_i \mid \mathbf{pa}_i^{\mathcal{G}})$ estimated separately in the environments.*

This phenomenon also occurs in undirected probabilistic graphical models, where the joint density $p(x_1, \ldots, x_n) = \frac{1}{Z} \prod_{c \in \text{cliques}} \Psi_c(x_c)$ is recoverable given potentials learnt from environments that contain the variables appearing in each clique. The above are the simplest cases of OOV generalization, yet they already illustrate that causal assumptions can help. We will study a more subtle case below.

To this end, we model an environment $\mathcal{E} = (\mathcal{D}, \mathcal{T})$ as a domain $\mathcal{D}$ and a task $\mathcal{T}$. The domain contains a variable space $\mathcal{X} := (X_1, X_2, \ldots)$ and its joint probability distribution $P(\mathcal{X})$. Given a domain, a task contains a target variable space $\mathcal{Y} := Y$, and a predictor $f : \mathcal{X} \to \mathcal{Y}$. To differentiate between components belonging to the source and target environments, subscripts $s$ and $t$ are used.

**Definition 1** (OOV Generalization). *OOV uncertainty arises when the variable space of the target environment is not contained in any of the variable spaces of the source environments, i.e., $\forall s, \{\mathcal{X}_t, \mathcal{Y}_t\} \not\subseteq \{\mathcal{X}_s, \mathcal{Y}_s\}$. If a method for estimating a quantity in the target environment (e.g., a predictor $f_t$) improves by utilizing data from the source environments, we say it generalizes OOV.*

While there is nothing causal about OOV generalization, we use SCMs since it turns out that they allow the formulation of assumptions and methods that provably exhibit OOV generalization.

## 3 RESIDUAL GENERALIZATION UNDER CAUSAL ASSUMPTIONS

### 3.1 PROBLEM FORMULATION

For simplicity, we consider a univariate setting, referring to Appendix C.1 for a multivariate extension. Consider an SCM with additive noise (Hoyer et al., 2009)

$$Y := \phi(X_1, X_2, X_3) + \epsilon \tag{3}$$

with a function $\phi$, jointly independent causes $X_i$, and $\epsilon \sim \mathcal{N}(0, \sigma^2)$. Assume that we do not have access to an environment jointly containing all variables $(X_1, X_2, X_3, Y)$. Instead, we have (Fig. 1a):

- A source environment with jointly observed $(X_1, X_2, Y)$, and
- A target environment with jointly observed $(X_2, X_3)$, and unobserved $Y$.

Our goal is to predict $Y$ given $X_2, X_3$.

This OOV scenario posits two challenges: without joint observations of $(X_2, X_3, Y)$, we cannot train or fine-tune a discriminative model in the target environment; further, due to the independence among the covariates, it is impossible to infer $X_3$ from the covariates observed in the source environment. Fig. 2a shows a visualization of the problem.

To ground the problem in the real world, consider two medical labs collecting different sets of variables. Lab A collects $X_1 = $ lifestyle factors and $X_2 = $ blood test; Lab B, in addition to $X_2$, collects $X_3 = $ genomics. Lab A is hospital-based and can measure diseases $Y$, whereas B is a research lab. The OOV problem asks: given a model trained to predict $Y$ on Lab A's data, how should Lab B use this model for its own dataset that differs in the set of input variables?

### 3.2 CHALLENGES IN DISCRIMINATIVE OOV GENERALIZATION

Transfer learning often transfers reusable features or model parameters of a discriminative model fitted in the source environment. This approach has inherent limitations when it comes to OOV scenarios: lacking the outcome variable $Y$, we cannot fine-tune in the target environment; further the common features, in our case, are the common variables $X_2$ shared between environments. A naive approach is then to predict the target sample using the model restricted to $X_2$, i.e., the *marginal predictor*, cf. our experiments (§ 4). However, such a predictor yields constant prediction irrespective of changing $X_3$.

To leverage the shared variables further, we study the properties that the optimal target predictor is expected to satisfy to restrict the set of potential target predictors. To see this concretely: suppose data is sufficient and one can observe all variables of interest, training discriminative models on each environment yields the optimal predictive functions that minimize the mean squared error loss

$$f_s(x_1, x_2) = \mathbb{E}_{X_3}[Y \mid x_1, x_2] \tag{4}$$

$$f_t(x_2, x_3) = \mathbb{E}_{X_1}[Y \mid x_2, x_3] \tag{5}$$

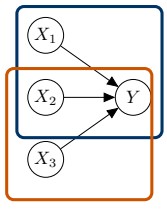 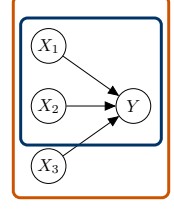 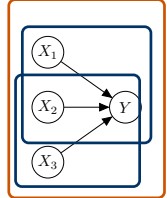

(a) Marginal to Marginal     (b) Marginal to Joint     (c) Merge datasets

Figure 2: Examples of OOV scenarios where marginal consistency condition alone (6) does not permit the identification of the optimal predictive function in the corresponding target domain.

for the source and target environment, respectively. With the discriminative model $f_s$ fitted in the source environment, its residual distribution is the distribution of differences between the observed value and the prediction, $Y - f_s(X_1, X_2) \mid X_1, X_2$. Note that the optimal predictive functions $f_s, f_t$ automatically satisfy the **marginal consistency** condition (see Appendix A): for any $x_2$, we have

$$\mathbb{E}_{X_1}[f_s(X_1, x_2)] = \mathbb{E}_{X_3}[f_t(x_2, X_3)] \tag{6}$$

Suppose we have trained the optimal predictor in the source environment. The marginal consistency condition (6) enforcing consistency over the shared variables, then restricts the solution space for the predictor in the target environment. However, Theorem 1 shows that this restriction does not uniquely determine the target predictor, i.e., it does not permit *identification* of the optimal predictive function in the target environment for all the scenarios shown in Fig. 2. See Appendix C.2 for the multivariate version, and proofs.

**Theorem 1.** *Consider the OOV scenarios in Fig. 2, each governed by the SCM described in § 3.1. Suppose that the variables considered in Fig. 2a and Fig. 2b are real-valued and the variables $X_1$ and $X_3$ in Fig. 2c are binary. We assume that for all $i$, the marginal density $p_i(x_i)$ is known, and denote its support set as $S_i := \{x \in \mathbb{R} \mid p_i(x) > 0\}$. Suppose that for all $i$ there exist two distinct points $x, x' \in S_i$. Then, for any pair $f_s, f_t$ satisfying marginal consistency (6) and for any $R > 0$, there exists another function $f_t'$ with $\|f_t - f_t'\|_2 \geq R$ that also satisfies marginal consistency.*

### 3.3 IDENTIFICATION IN OOV GENERALIZATION

We now study when identifiability of the optimal target predictive function can be achieved. To start with we consider a different setting than our setup (Fig. 1), where covariates between source and target environments are independent and causes of the outcome variable contained in the target environment.

#### 3.3.1 WITH DEPENDENT COVARIATES

**Theorem 2.** *Consider a target variable $Y$ and its direct cause $PA_Y$. Suppose that we observe:*

- *source environment contains variables $(Z, Y)$; training a discriminative model on this environment yields a function $f_s(z) = \mathbb{E}[Y \mid Z]$,*

- *target environment contains variable $PA_Y$*

*Suppose $Y := \phi(PA_Y) + \epsilon_Y$, $Z = g(PA_Y) + \epsilon_Z$ where $g$ is known and invertible with $\phi, g^{-1}$ uniformly continuous. Then in the limit of $\mathbb{E}[\|\epsilon_Z\|] \to 0$, the composition of the discriminative models in source environments also approaches the optimal predictor, i.e., $\forall pa_Y : f_s \circ g(pa_Y) \to \phi(pa_Y)$.*

Appendix C.3 details its multivariate statement and proof and Fig. 5 in Appendix shows an example of such a scenario. Informally, Theorem 2 states that one can identify the optimal target predictive function from the learnt source function in our setup if the dependence structure between the source and target covariates is known and satisfies the above assumptions. However, in real-world applications, the dependence structure between the source and target covariates may not be known or even exist. To further understand this OOV problem, we next study a more challenging scenario when all covariates are independent from each other, and demonstrate a seemingly surprising result, that under certain assumptions, the optimal target predictive function is identifiable without the knowledge of the dependence structure among covariates.

### 3.3.2 WITH INDEPENDENT COVARIATES

With theoretical results on the limitations of current approaches in transferring with discriminative model for OOV scenarios, we present a practical method for the base case illustrated in Fig. 1a and detail its underlying assumptions.

**Simple Additive Model** One solution to tackle the problem in Fig. 1a is to train separate discriminative models for each observed variable. For example, given the source environment, we learn function mappings on $(X_1, Y)$ and $(X_2, Y)$ as $f_1, f_2$. When facing a different set of variables, e.g., $(X_2, X_3)$, we could directly re-use the learnt $f_2$. With additional collection of $Y$ in the target domain, we then train a model on $(X_3, Y)$. The method offers a degree of compositional flexibility and circumvents the need to jointly observe variables of interest, e.g., $(X_2, X_3, Y)$. However, such a method first requires the collection of variable $Y$ in the target domain and assumes that the generating function of $Y$ has only linear relationships with its causes (i.e., there is no interaction term like $X_i X_j, i \neq j$). A detailed description of the model and its underlying assumptions can be found in Appendix B.

Below, we propose a method to transfer in OOV scenarios that 1) does not require us to observe $Y$ in the target domain, and 2) relaxes the linearity assumption. Note the main idea is general to work for all scenarios in Fig. 2. We illustrate our method via an example (§ 3.3.3) with details in § 3.3.4.

### 3.3.3 MOTIVATING EXAMPLE

Consider the problem described in § 3.1 in the case where $\phi$ is a polynomial:

$$Y := \alpha_1 X_1 + \alpha_2 X_2 + \alpha_3 X_3 + \alpha_4 X_1 X_2 + \alpha_5 X_1 X_3 + \alpha_6 X_2 X_3 + \alpha_7 X_1 X_2 X_3 + \epsilon \quad (7)$$

Let $X_i$ have mean $\mu_i$, variance $\sigma_i$ for all $i$. Given sufficient data and the observation of variable $Y$ in the target environment, we train discriminative models in each environment, yielding the optimal predictive functions that minimize the mean squared error as:

$$\begin{aligned} f_s(x_1, x_2) &= (\alpha_3 \mu_3) + (\alpha_1 + \alpha_5 \mu_3)x_1 + (\alpha_2 + \alpha_6 \mu_3)x_2 + (\alpha_4 + \alpha_7 \mu_3)x_1 x_2 \\ f_t(x_2, x_3) &= (\alpha_1 \mu_1) + (\alpha_3 + \alpha_5 \mu_1)x_3 + (\alpha_2 + \alpha_4 \mu_1)x_2 + (\alpha_6 + \alpha_7 \mu_1)x_2 x_3 \end{aligned} \quad (8)$$

We first illustrate fine-tuning, in this example, cannot identify target predictive function. Note coefficients $\{\alpha_i\}$ are model parameters. We observe the coefficients for the common term $x_2$ share some constituents between $f_s$ and $f_t$ in (8). One can thus expect, during fine-tuning, the coefficients may adapt quickly. However, it is clear that one cannot uniquely determine the coefficients of $f_t$ without observing $Y$ from the target environment, since the system of equations is under-determined with eight unknown coefficients and four estimated values – even in the above polynomial case.

**'No noise' regime** First assume that there is no noise, i.e. $\epsilon = 0$. Although we do not observe the cause $X_3$, we nevertheless have information about it in the source environment: The unobserved variable act as a noise term, and the residual distribution in the source environment carries a footprint of it. We will see below that subject to suitable assumptions, this idea carries over to the noisy case.

**'With noise' regime** Now consider additional additive noise. We will see that the idea outlined above carries over under suitable assumptions. The third moment from the residual distribution in the source environment takes the following form:

$$\mathbb{E}\big[(Y - f_s(x_1, x_2))^3 \mid x_1, x_2\big] = (\alpha_3 + \alpha_5 x_1 + \alpha_6 x_2 + \alpha_7 x_1 x_2)^3 \, \mathbb{E}[(X_3 - \mu_3)^3] \quad (9)$$

We observe that the term in parentheses coincides exactly with the partial derivative, i.e.,

$$\left. \frac{\partial \phi}{\partial X_3} \right|_{x_1, x_2, \mu_3} = \alpha_3 + \alpha_5 x_1 + \alpha_6 x_2 + \alpha_7 x_1 x_2 \quad (10)$$

Under $\phi$ in (7) as a polynomial, we know terms in the source environment with non-zero coefficients are $g(x_1, x_2) = [1, x_1, x_2, x_1 x_2]$. One can then fit a linear model with features in $g$ on the source environment and estimate the coefficients. The resulting predictor is $f_s(x_1, x_2) = \beta^T g(x_1, x_2)$, where $\beta_1 = \alpha_3 \mu_3, \beta_2 = \alpha_2 + \alpha_5 \mu_3, \beta_3 = \alpha_2 + \alpha_6 \mu_3, \beta_4 = \alpha_4 + \alpha_7 \mu_3$. It is clear that learning $\beta$ alone cannot uniquely determine the coefficients $\alpha_i$. This intuition is supported by Theorem 1. To illustrate the main idea of our method, consider the error in the source environment after fitting a linear predictive model $f_s$: $Y - f_s(x_1, x_2)$. Let $W$ be some transformation of the error, where

$W = (Y - f_s(x_1, x_2))^3/k_3$ and $k_3 = \mathbb{E}[(X_3 - \mu_3)^3]$ estimated by observed $X_3$ samples in the target environment. Fit $W$ against $(\theta^T g(x_1, x_2))^3$ and estimate the coefficients $\theta$, as shown in (9), enables the estimation of the coefficients $\alpha_3, \alpha_5, \alpha_6, \alpha_7$. Combined with the estimated coefficients $\beta$, we can uniquely determine the coefficients of $f_t$ without the need to observe $Y$ from the target environment.

**Discussion** The intuition behind this seemingly surprising result is rather straightforward – $X_3$ though unobserved in the source, is a generating factor of $Y$. Its information is not only contained in the marginalized mean but also in the residual distribution of the error after fitting a discriminative model.

### 3.3.4 OUT-OF-VARIABLE LEARNING

This phenomenon is extendable to more general settings. Theorem 3 shows that the moments in the residuals still provide additional information about the partial derivative of the function $\phi$ w.r.t $X_3$ for general nonlinear smooth functions. Appendix C.4 shows its multivariate statement and the proof.

**Theorem 3.** *Consider the problem setup in § 3.1 and assume the function $\phi$ is everywhere twice differentiable with respect to $X_3$. Suppose from the source environment we learn a function $f_s(x_1, x_2) = \mathbb{E}[Y \mid x_1, x_2]$. Using first-order Taylor approximation on the function $\phi : x_1 \times x_2 \times \mathcal{X}_3 \to \mathbb{R}$ for fixed $x_1, x_2$, the moments of the residual distribution in the source environment take the form*

$$\mathbb{E}[(Y - f_s(x_1, x_2))^n \mid x_1, x_2] = \sum_{k=0}^{n} \binom{n}{k} \mathbb{E}[\epsilon^k] \left( \left. \frac{\partial \phi}{\partial X_3} \right|_{x_1, x_2, \mu_3} \right)^{n-k} \mathbb{E}[(X_3 - \mu_3)^{n-k}]. \quad (11)$$

*For $n = 3$, this reduces to*

$$\mathbb{E}[(Y - f_s(x_1, x_2))^3 \mid x_1, x_2] = \left( \left. \frac{\partial \phi}{\partial X_3} \right|_{x_1, x_2, \mu_3} \right)^3 \mathbb{E}[(X_3 - \mu_3)^3] + \mathbb{E}[\epsilon^3]. \quad (12)$$

Theorem 3 shows that the moments of the residual distribution include a contribution from both the moments of the noise variable and the propagated effects caused by variables unique to the target environment. When $n = 3$, most terms that involve the undesired noise variable disappear.

**Corollary 4.** *For OOV scenarios described in § 3.1, learning from the moment of the error distribution allows exact identification of $\phi$ when $\phi(x_1, x_2) = \sum_{p,q} c_i h(x_1, x_2)^p x_3^q$, where $p, q \in \{0, 1\}$ and $c_i \in \mathbb{R}, \forall i$ and $h$ can be any function.*

To see Corollary 4 in action, recall when $\phi$ is as in (7), our solution is analytically exact, as shown in (10). Theorem 3 and Corollary 4 demonstrate that in this challenging OOV scenario where existing transfer learning methods fail to apply (cf. Theorem 1), learning from the residual distribution offers exact identification for a certain class of generating functions.

Next, we build a practical predictor that utilizes the above theoretical insights and present experimental results to evaluate OOV learning performance. To start with, note the target predictive function can be Monte Carlo approximated if the true function $\phi$ is known:

$$f_t(x_2, x_3) = \int \phi(x_1, x_2, x_3) p(x_1) dx_1 \approx \frac{1}{n} \sum_{i=1}^{n} \phi(x_{1,i}, x_2, x_3), \quad \text{where } x_{1,i} \sim p(x_1)$$

Assume $\phi$ is smooth, by first-order Taylor approximation evaluated at $(x_1, x_2, \mu_3)$, rewrite $\phi$ as:

$$\phi(x_1, x_2, x_3) = \phi(x_1, x_2, \mu_3) + \left. \frac{\partial \phi}{\partial X_3} \right|_{x_1, x_2, \mu_3} (x_3 - \mu_3) + \mathcal{O}((x_3 - \mu_3)^2) \quad (13)$$

Taking expectations of $X_3$ on both sides of Eq. 13, we see that $f_s(x_1, x_2) \approx \phi(x_1, x_2, \mu_3)$. Theorem 3 states that we can estimate the partial derivative term from the third moment of the error distribution. We thus propose *MomentLearn*, an OOV estimate $\tilde{f}_t$ for the target predictive function:

$$\tilde{f}_t(x_2, x_3) = \frac{1}{n} \sum_{i=1}^{n} f_s(x_{1,i}, x_2) + h_\theta(x_{1,i}, x_2)(x_3 - \mu_3), \quad \text{with } x_{1,i} \sim p(x_1), \quad (14)$$

where $h_\theta$ is a MLP parameterized by $\theta$, modelling the partial derivative by regressing on the 3rd moment of the residual distribution from the source. Note the proposed predictor is strictly better than naïvely marginalizing $f_s$ on the shared variables. The proposed estimate also satisfies the marginal consistency condition as the second term in (14) vanishes when taking expectations. See Algorithm 1 in Appendix D.3 for a detailed procedure.

|            | $\sum_i \alpha_i X_i$ | $+\sum_{i<j} \beta_{ij} X_i X_j$ | $+\sum_i \gamma_i X_i^2$ |
|------------|-----------------------|----------------------------------|--------------------------|
| Oracle     | $0.31 \pm 0.15$       | $0.26 \pm 0.44$                  | $0.57 \pm 0.32$          |
| MomentLearn| $0.32 \pm 0.15$       | $0.31 \pm 0.48$                  | $0.71 \pm 0.37$          |
| MeanImputed| $0.45 \pm 0.21$       | $0.48 \pm 0.45$                  | $1.50 \pm 1.03$          |
| Marginal   | $0.52 \pm 0.45$       | $0.80 \pm 0.75$                  | $1.75 \pm 1.27$          |

|            | $\mathcal{GP}_1(X_1, X_2) + \alpha_3 X_3$ | $+\mathcal{GP}_2(X_1, X_2) \cdot X_3$ | $+\mathcal{GP}_3(X_1, X_2) \cdot X_3^2$ |
|------------|-------------------------------------------|----------------------------------------|------------------------------------------|
| Oracle     | $0.06 \pm 0.04$                           | $0.08 \pm 0.06$                        | $0.18 \pm 0.14$                          |
| MomentLearn| $0.06 \pm 0.04$                           | $0.10 \pm 0.06$                        | $0.67 \pm 0.59$                          |
| MeanImputed| $0.37 \pm 0.26$                           | $0.32 \pm 0.25$                        | $1.31 \pm 0.95$                          |
| Marginal   | $0.33 \pm 0.14$                           | $0.41 \pm 0.37$                        | $1.46 \pm 1.21$                          |

Table 1: Our method's ("MomentLearn") OOV prediction performance in the target environment, compared to the "Marginal" baseline, the predictor that imputes missing variable with its mean ("Mean Imputed") and the solution that has access to the full joint observations on the target domain ("Oracle"). Shown are mean and standard deviations of the MSE loss between the predicted and observed target values. $\mathcal{GP}_i(\cdot)$ denotes a function sampled from a Gaussian Process with zero mean and Gaussian kernel. MomentLearn performs as expected by our theoretical results and even exhibits a degree of robustness to function classes that are not covered by Theorem 3.

## 4 EXPERIMENTS

We perform both synthetic and real-world (Appendix D.4) experiments to evaluate our algorithm's OOV learning performance. *OOV* in this context means that $X_3$ is not observed in the source, and we do not observe $Y$ in the target environment. We generate synthetic data according to § 3.1 for a range of function classes. The inputs $\mathbf{X} \in \mathbb{R}^3$ are independently generated from a Gamma distribution. Variable $Y$ is a function of the inputs, and the observed values are generated with noise, $Y_{\text{obs}} = Y + \epsilon$, where $\epsilon \sim \mathcal{N}(0, \sigma^2), \sigma = 0.1$. We benchmark our method's performance against several baselines. As a measure of performance against an *oracle* solution, we compare with the predictor trained from scratch if we jointly observe *all* variables in the target environment on large data sets. To highlight the need to predict beyond marginalizing learnt models on the common variable, we compare with the *marginal* predictor. To benchmark against mean imputation method for missing data problem, we compare with the *mean imputed* predictor. See Appendix D.1 for implementation details and Appendix E.2.1 on robustness analysis with respect to hyperparameters (D.2), varying noise scales (D.5) and heavy-tailed noise distribution (D.6).

**Prediction Performance** To evaluate our method's performance, we compare contour plots of prediction on the target variable $Y$ given covariates $X_2, X_3$ in the target environment. Its functional relationship is as described in Eq. 7. Fig. 1 (b)-(e) shows that our method's solution is almost identical to the oracle solution, whereas the marginal and mean imputed predictor deviate far from the oracle.

**Systematic Analysis** To systematically analyse the robustness of our method w.r.t different function classes, we compare our method's prediction against the marginal, mean imputed and oracle predictor on increasingly more complex functions. We start with base functions with linear additive terms, i.e., $f_1(X_1, X_2, X_3) = \sum_i \alpha_i X_i$. In addition to base functions, we consider functions that additionally incorporate linear interaction terms, i.e., $f_2(X_1, X_2, X_3) = f_1(X_1, X_2, X_3) + \sum_{i<j} \beta_{ij} X_i X_j$. For the final function class, we incorporate additional square terms $f_3(X_1, X_2, X_3) = f_2(X_1, X_2, X_3) + \sum_i \gamma_i X_i^2$. We use $10k$ data in the source environment and randomly sample 5 functions in each function class and average the results after a hyperparameter sweep. Table 1 records the mean and standarad deviation of the MSE loss between the predicted and observed target values for different methods. We observe the proposed MomentLearn performs comparatively with the oracle predictor and consistently outperforms both the marginal and mean imputed predictors in its accuracy and reliability even for the function class $f_3$, which the method is not guaranteed to identify. We further evaluate more general functions, where we sample 5 functions randomly generated by Gaussian processes with Gaussian kernel. Just as above, we observe a similar result which supports our theoretical results.

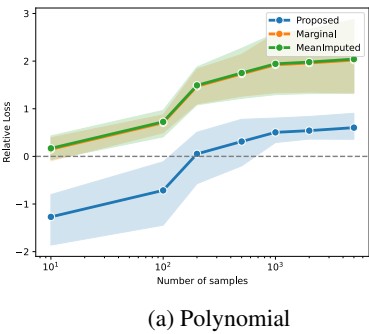
(a) Polynomial

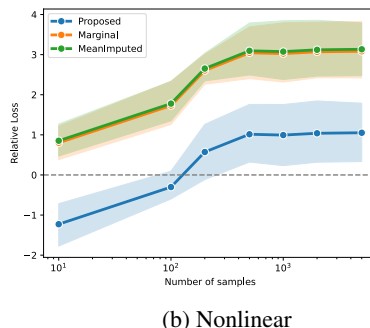
(b) Nonlinear

Figure 3: Shown are mean of the relative loss (and its $95\%$ confidence interval) for varying numbr of joint samples observed in the target domain. MomentLearn outperforms the joint predictor in the few-sample region and is always preferred over the marginal predictor.

**Sample Efficiency** To evaluate our method's comparative advantage in terms of sample efficiency had we observed $Y$ in the target domain, we consider a few-shot learning setting. Suppose we observe a few samples with joint variables $(X_2, X_3, Y)$ in the target environment. We generate the target variable $Y$ as a function of the inputs, where the function takes either polynomial or nonlinear form: $Y_{poly} = \boldsymbol{\alpha}^T \text{Poly}(\tilde{\mathbf{X}}), Y_{nonlinear} = \sqrt{(\boldsymbol{\alpha}^T \tilde{\mathbf{X}})^{\circ 2}}$, where $\tilde{\mathbf{X}}$ are standardized covariates, $\text{Poly}(\tilde{\mathbf{X}})$ are polynomial features as in Eq. 7, the coefficients $\boldsymbol{\alpha} \sim \mathcal{N}(\mathbf{0}, \mathbf{I})$ and $\circ 2$ denotes elementwise square operation. Fig.3 shows the mean of the relative loss (and its $95\%$ confidence interval) over 5 runs, $\log(\text{loss}/\text{loss}_o)$, where loss is the OOV loss of the respective method trained on $100k$ points from the source environment, and $\text{loss}_o$ is the loss of the joint predictor trained with varying numbers of joint observations in the target environment. When the relative loss is zero (dashed black line), the method is on par with the oracle predictor. When the relative loss is below zero, the respective method achieves a lower loss than the joint predictor trained on actual joint samples. We see that the proposed method always outperforms the marginal and mean imputed predictors. Both predictors never outperform training from scratch, irrespective of the number of joint samples observed. The proposed method beats the joint predictor until about 100 joint points are used.

## 5 CONCLUSION

We used Fig. 1 to suggest that it would seem hard to enable transfer from the source environment (blue box) to the target environment (orange box). We supported this intuition by Theorem 1. However, we also showed that under certain assumptions (e.g., the variables follow a causal graph with additive noise, and the functions in the data generation process are smooth), there is a valid source of information in this OOV scenario, enabling exact identification in certain function classes. We also proposed an algorithm to utilize this information and showed experiments in which the algorithm exhibited a degree of robustness with respect to a violation of the theoretical conditions.

We only considered the error of a target predictive function as a performance condition in Def. 1. We note that in the field of OOD generalization, a larger variety of settings has been considered (Wildberger et al., 2023). We briefly discuss the extension of OOV in multi-environments (E.1), its robustness (E.2.1) and its potential applications (E.3). Clearly, real-world problems require systems to achieve both OOD and OOV generalization. AI in medicine, for example, requires us to tackle the OOV problem for rare disease prediction and sample-efficient generalization.

We are far from being able to claim robust methods for real-world practical problems — the present contribution lies mainly in opening up avenues for future research. Some of them are of a conceptual nature, and some connected to the limitations of the present approach. We consider this work conceptually novel, exploring how generalization is intricately related to observability of variables and their (causal) relationships. We hope our work inspires further studies to explore and develop methodologies that apply to different OOV problems.

ACKNOWLEDGMENT

B.S. would like to acknowledge a number of discussions with Dominik Janzing during the last decades that helped him understand the role of additional variables in causal modeling. S.G. would like to thank Vincent Berenz for his valuable contribution in systematizing the codeing framework for broader usage.

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
