## A    Marginal Consistency

Let $h : \mathcal{X} \to \mathcal{Y}$ be a function defined in the joint environment $J$, where $\mathcal{X} := (X_1, X_2, \ldots)$ is a variable space. Let $S$ be a set and $S^c$ its complement. Denote $\mathcal{X}_S := \{X_i : i \in S\}$ to be the set of variables contained in the set $S$. $h_S$ is the function $h$ restricted to the set of variables in $\mathcal{X}_S$, i.e. $h_S : \mathcal{X}_S \to \mathcal{Y}$, where $h_S(x_S) = \mathbb{E}_{X_{S^c}}[h(x_S, X_{S^c})]$. Similarly, for $h_T : \mathcal{X}_T \to \mathcal{Y}$, where $h_T(x_T) = \mathbb{E}_{X_{T^c}}[h(x_T, X_{T^c})]$.

When $S \cap T \neq \emptyset$, let $I := S \cap T$, then $h_I : \mathcal{X}_I \to \mathcal{Y}$, where $h_I(x_I) = \mathbb{E}_{X_{I^c}}[h(x_I, X_{I^c})]$. Since $I \subseteq S, S^c \subseteq I^c$, define $A_S = I^c \setminus S^c$ and similarly $B_T = I^c \setminus T^c$. Then,

$$h_I(x_I) = \mathbb{E}_{X_{I^c}}[h(x_I, X_{I^c})] \tag{15}$$

$$= \mathbb{E}_{X_{A_S}}[\mathbb{E}_{X_{S^c}}[h(x_S, X_{S^c})]] \tag{16}$$

$$= \mathbb{E}_{X_{B_T}}[\mathbb{E}_{X_{T^c}}[h(x_T, X_{T^c})]] \tag{17}$$

$$= \mathbb{E}_{X_{A_S}}[h_S(x_I, X_{A_S})] = \mathbb{E}_{X_{B_T}}[h_T(x_I, X_{B_T})] \tag{18}$$

## B    Naïve Model

For simplicity, we consider the structural causal model described in § 3.1. The method described below also applies if we replace variable $X_i$ by any subsets of variables. Assume that the true generating function $\phi$ is additively composed of univariate functions of its covariates, i.e. there is no mixing term between different covariates. Then the additive model is given as:

$$Y := f_1(X_1) + f_2(X_2) + f_3(X_3) + \epsilon \tag{19}$$

where $X_1, X_2, X_3$ are jointly independent of each other and $\epsilon \sim \mathcal{N}(0, \sigma^2)$. Further, $f_1, f_2, f_3$ are some unknown functions.

We consider the following two environments:

- The source environment $\mathcal{E}_S$ contains variables $(X_1, X_2, Y)$
- The target environment $\mathcal{E}_t$ contains variables $(X_2, X_3, Y)$

The goal is to transfer knowledge from environment $\mathcal{E}_S$ to environment $\mathcal{E}_t$ in order to learn the predictor $\mathbb{E}[Y \mid X_2, X_3]$.

One approach is to build separate neural networks for datasets $(X_1, Y)$ and $(X_2, Y)$. With sufficient data, the learnt functions will equals $f_1, f_2$ respectively due to independence between $X_i$. Knowledge transfer can the be achieved by migrating the learnt function $f_2$ to the target environment and then using data from the target environment to learn the function $f_3$ over the previously unobserved variable. This solution does not work if there is an interaction between covariates, since the learnt function does not solely depend on the unobserved variable.

## C    Proofs

We next show proofs for theorems in the main text. We will first state theorem in its single-variate form and state its multivariate extension. As single-variate proof is easily deducible from its multivariate extension, we will only show its multivariate proof. To start, we will formulate an equivalent problem setup for multivariate cases.

### C.1    Problem Formulation in Multivariate Version

Consider a simple structural causal model with additive noise (Hoyer et al., 2009):

$$Y := \phi(\mathbf{PA}_Y) + \epsilon \tag{20}$$

where $\phi$ is some function, $Y \in \mathbb{R}$ and $\epsilon \sim \mathcal{N}(0, \sigma^2)$ and $X_i \in \mathbf{PA}_Y$ are jointly independent causes. Assume that we do not have access to a joint environment $J$ that contains all variables of interest, namely $(\mathbf{PA}_Y, Y)$. Instead, we have:

- A source environment with observed variables $(\mathbf{X}_s, Y)$, and
- A target environment with observed variables $\mathbf{X}_t$ and the (unobserved) variable $Y$.

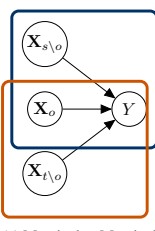 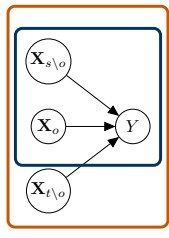 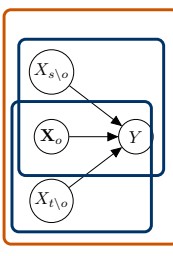

(a) Marginal to Marginal      (b) Marginal to Joint      (c) Merge datasets

Figure 4: Examples of OOV scenarios where marginal consistency condition alone (6) does not permit the identification of the optimal predictive function in the corresponding target domain.

## C.2 THEOREM 1

### C.2.1 SINGLE-VARIATE STATEMENT

**Theorem 5.** *Consider the OOV scenarios in Fig. 2, each governed by the SCM described in § 3.1. Suppose that the variables considered in Fig. 2a and Fig. 2b are real-valued and the variables $X_1$ and $X_3$ in Fig. 2c are binary. We assume that for all $i$, the marginal density $p_i(x_i)$ is known, and denote its support set as $S_i := \{x \in \mathbb{R} \mid p_i(x) > 0\}$. Suppose that for all $i$ there exist two distinct points $x, x' \in S_i$. Then, for any pair $f_s, f_t$ satisfying marginal consistency (6) and for any $R > 0$, there exists another function $f'_t$ with $\|f_t - f'_t\|_2 \geq R$ that also satisfies marginal consistency.*

### C.2.2 MULTIVARIATE STATEMENT

**Theorem 5.** *Consider the OOV scenarios illustrated in Fig. 4, each governed by the structural causal model described in appendix C.1. Suppose that the variables considered in Fig. 4a and Fig. 4b are real-valued and the variables $X_{s\backslash o}$ and $X_{t\backslash o}$ in Fig. 4c are binary. We assume that for all $i \in \{s\backslash o, o, t\backslash o\}$, the marginal density $p_i(\mathbf{x}_i)$ is known, and denote its support set as $S_i := \{\mathbf{x} \in \mathbb{R}^n \mid p_i(\mathbf{x}) > 0\}$. Suppose that for all $i$ there exist two distinct points $\mathbf{x}, \mathbf{x}' \in S_i$. Then, for any pair $f_s, f_t$ satisfying marginal consistency (6) and for any $R > 0$, there exists another function $f'_t$ with $\|f_t - f'_t\|_2 \geq R$ that also satisfies marginal consistency.*

### C.2.3 PROOF

*Proof.* Proof by construction. Consider the scenario illustrated in Fig. 4b. Find two distinct values in the support set of $p_{t\backslash o}(\mathbf{x}_{t\backslash o})$, $\mathbf{x}'_{t\backslash o}$ and $\mathbf{x}''_{t\backslash o}$. For some appropriate $\epsilon > 0$, consider their neighbourhoods as $N_1 = [\mathbf{x}'_{t\backslash o} - \epsilon, \mathbf{x}'_{t\backslash o} + \epsilon]$ and $N_2 = [\mathbf{x}''_{t\backslash o} - \epsilon, \mathbf{x}''_{t\backslash o} + \epsilon]$ and $N_1 \cap N_2 = \emptyset$. Suppose we learnt the optimal predictive function in the source environment $f_s$, it can be written as:

$$f_s(\mathbf{x}_{s\backslash o}, \mathbf{x}_o) = \int_\Omega \phi(\mathbf{x}_{s\backslash o}, \mathbf{x}_o, \mathbf{x}_{t\backslash o}) p_{t\backslash o}(\mathbf{x}_{t\backslash o}) d(\mathbf{x}_{t\backslash o}) \tag{21}$$

$$= \underbrace{\int_{(\Omega \backslash N_1) \backslash N_2} \phi(\mathbf{x}_{s\backslash o}, \mathbf{x}_o, \mathbf{x}_{t\backslash o}) p_{t\backslash o}(\mathbf{x}_{t\backslash o}) d(\mathbf{x}_{t\backslash o})}_{\text{Remainder}(\mathbf{x}_{s\backslash o}, \mathbf{x}_o)} \tag{22}$$

$$+ \underbrace{\int_{N_1} \phi(\mathbf{x}_{s\backslash o}, \mathbf{x}_o, \mathbf{x}_{t\backslash o}) p_{t\backslash o}(\mathbf{x}_{t\backslash o}) d(\mathbf{x}_{t\backslash o})}_{g(\mathbf{x}_{s\backslash o}, \mathbf{x}_o)} + \underbrace{\int_{N_2} \phi(\mathbf{x}_{s\backslash o}, \mathbf{x}_o, \mathbf{x}_{t\backslash o}) p_{t\backslash o}(\mathbf{x}_{t\backslash o}) d(\mathbf{x}_{t\backslash o})}_{h(\mathbf{x}_{s\backslash o}, \mathbf{x}_o)}$$
$$\tag{23}$$

Denote the integral in the region excluding the specified neighbourhoods as $\text{Remainder}(\mathbf{x}_{s\backslash o}, \mathbf{x}_o)$, the integral over $N_1$ as $g(\mathbf{x}_{s\backslash o}, \mathbf{x}_o)$, and that over $N_2$ as $h(\mathbf{x}_{s\backslash o}, \mathbf{x}_o)$. Given any function $c(\mathbf{x}_{s\backslash o}, \mathbf{x}_o)$, it is easy to find a function $d(\mathbf{x}_{s\backslash o}, \mathbf{x}_o)$ such that $f_s(\mathbf{x}_{s\backslash o}, \mathbf{x}_o) - \text{Remainder}(\mathbf{x}_{s\backslash o}, \mathbf{x}_o) = c(\mathbf{x}_{s\backslash o}, \mathbf{x}_o)g(\mathbf{x}_{s\backslash o}, \mathbf{x}_o) + d(\mathbf{x}_{s\backslash o}, \mathbf{x}_o)h(\mathbf{x}_{s\backslash o}, \mathbf{x}_o)$. This means whenever we find a function $\phi$ that satisfies Equation 21, it is always possible to slightly perturb $\phi$ such that $\phi'$ can also satisfy marginal

consistency. For example, construct a $\phi'$ which is a result of proposed $\phi$ scaled by $c$ elementwise over the neighbourhood $N_1$ and scaled by $d$ elementwise over the neighbourhood $N_2$. Moreover, the deviation of $\phi'$ with $\phi$ can be arbitrarily different:

$$||\phi - \phi'||_2 \geq \text{const}_1 ||c - 1||_2 + \text{const}_2 ||d - 1||_2 \tag{24}$$

which the lower bound can be arbitrarily large by the choice of $c(x_1, x_2)$.

Consider the scenario illustrated in Fig. 4a. Given the information obtained from the source environment $p_{s\backslash o}(\mathbf{x}_{s\backslash o}), p_o(\mathbf{x}_o), f_s(\mathbf{x}_{s\backslash o}, \mathbf{x}_o)$, by argument above, we know it is always possible to perturb learnt $\phi$ appropriately to get $\phi'$ that satisfies the desired marginal consistency conditions. We will show it will also be impossible to identify the optimal predictive function $f_t$ in the target environment. By the argument above, we can choose the function $c(\mathbf{x}_{s\backslash o}, \mathbf{x}_o)$ freely. Then there always exists a function $c$ such that $\text{sgn}(c(\mathbf{x}_{s\backslash o}, \mathbf{x}_o)) = \text{sgn}(\phi(\mathbf{x}_{s\backslash o}, \mathbf{x}_o, \mathbf{x}'_{t\backslash o}))$ and $|c(\mathbf{x}_{s\backslash o}, \mathbf{x}_o)| \geq L, \forall \mathbf{x}_{s\backslash o}, \mathbf{x}_o$, where $L > 1$. Consider a point $\mathbf{x}'_{t\backslash o}$ in the neighbourhood $N_1$. Then under a learnt $\phi$ and perturbed $\phi'$, its corresponding optimal predictive function $f_t$ and $f'_t$ can be written as:

$$f_t(\mathbf{x}_o, \mathbf{x}'_{t\backslash o}) = \int_\Omega \phi(\mathbf{x}_{s\backslash o}, \mathbf{x}_o, \mathbf{x}'_{t\backslash o}) p_{s\backslash o}(\mathbf{x}_{s\backslash o}) d(\mathbf{x}_{s\backslash o}) \tag{25}$$

$$f'_t(\mathbf{x}_o, \mathbf{x}'_{t\backslash o}) = \int_\Omega \underbrace{c(\mathbf{x}_{s\backslash o}, \mathbf{x}_o)\phi(\mathbf{x}_{s\backslash o}, \mathbf{x}_o, \mathbf{x}'_{t\backslash o})}_{\geq 0 \text{ and } \neq \phi} p_{s\backslash o}(\mathbf{x}_{s\backslash o}) d(\mathbf{x}_{s\backslash o}) \tag{26}$$

This implies $f_t(\mathbf{x}_o, \mathbf{x}'_{t\backslash o}) \neq f'_t(\mathbf{x}_o, \mathbf{x}'_{t\backslash o})$ for all values in the neighbourhood $N_1$. This means though $f_t$ and $f'_t$ are both marginally consistent with $f_s$ (since $\phi$ and $\phi'$ are both consistent with $f_s$), but they are different functions. Moreover their difference can be arbitrarily large:

$$||f_t - f'_t||_2 \geq \int \int_{N_1} (f_t(\mathbf{x}_o, \mathbf{x}'_{t\backslash o}) - f'_t(\mathbf{x}_o, \mathbf{x}'_{t\backslash o}))^2 p_{t\backslash o}(\mathbf{x}_{t\backslash o}) d\mathbf{x}_{t\backslash o} p_o(\mathbf{x}_o) d\mathbf{x}_o \tag{27}$$

If analyse the inner term that squared, we have:

$$(f_t(\mathbf{x}_o, \mathbf{x}'_{t\backslash o}) - f'_t(\mathbf{x}_o, \mathbf{x}'_{t\backslash o}))^2 = (\int_\Omega (c-1)\phi(\mathbf{x}_{s\backslash o}, \mathbf{x}_o, \mathbf{x}_{t\backslash o}) p_{s\backslash o}(\mathbf{x}_{s\backslash o}) d\mathbf{x}_{s\backslash o})^2$$

$$\geq (|L| - 1)^2 (\int_\Omega \phi p_{s\backslash o}(\mathbf{x}_{s\backslash o}) d\mathbf{x}_{s\backslash o})^2$$

The last inequality holds by construction of $c$. Thus substituting it into Eq. 27, we have $||f_t - f'_t||_2 \geq \text{const} * (|L| - 1)^2$, where the lower bound of the constructed function $c$ can be arbitrarily large.

Consider the scenario illustrated in Fig. 4c. Here we restrict to cases when $X_{s\backslash o}$ and $X_{t\backslash o}$ contains singleton binary variables. For ease of notation, denote $X_{s\backslash o}$ as $X_1$, $X_{t\backslash o}$ as $X_3$ and $\mathbf{X}_o$ as $\mathbf{X}_2$. Set $\gamma_i := P(X_i = 0)$. Then given source environments where one observes variables $(X_1, \mathbf{X}_2, Y)$ and the other observes variables $(\mathbf{X}_2, X_3, Y)$. The potential generating function must satisfy below system of equations:

$$f_s(0, \mathbf{x}_2) = \gamma_3 \phi(0, \mathbf{x}_2, 0) + (1 - \gamma_3)\phi(0, \mathbf{x}_2, 1) \tag{28}$$
$$f_s(1, \mathbf{x}_2) = \gamma_3 \phi(1, \mathbf{x}_2, 0) + (1 - \gamma_3)\phi(1, \mathbf{x}_2, 1) \tag{29}$$
$$f_t(\mathbf{x}_2, 0) = \gamma_1 \phi(0, \mathbf{x}_2, 0) + (1 - \gamma_1)\phi(1, \mathbf{x}_2, 0) \tag{30}$$
$$f_t(\mathbf{x}_2, 1) = \gamma_1 \phi(0, \mathbf{x}_2, 1) + (1 - \gamma_1)\phi(1, \mathbf{x}_2, 1) \tag{31}$$

Perturb $\phi(0, \mathbf{x}_2, 0)$ by $c(0, \mathbf{x}_2, 0)$, then in order to still satisfy the above system of equations, the coefficients need to be correspondingly adjusted as:

$$c(0, \mathbf{x}_2, 1) = \frac{f_s(0, \mathbf{x}_2) - \gamma_3 c(0, \mathbf{x}_2, 0)\phi(0, \mathbf{x}_2, 0)}{(1 - \gamma_3)\phi(0, \mathbf{x}_2, 1)} \tag{32}$$

$$c(1, \mathbf{x}_2, 1) = \frac{f_t(\mathbf{x}_2, 1) - \frac{\gamma_1 f_t(0,\mathbf{x}_2) - \gamma_1 \gamma_3 c(0,\mathbf{x}_2,0)\phi(0,\mathbf{x}_2,0)}{(1-\gamma_3)}}{(1 - \gamma_1)\phi(1, \mathbf{x}_2, 1)} \tag{33}$$

$$c(1, \mathbf{x}_2, 0) = \frac{f_s(1, \mathbf{x}_2) - \frac{(1-\gamma_3)f_t(\mathbf{x}_2,1) - \gamma_1 f_s(0,\mathbf{x}_2) + \gamma_1 \gamma_3 c(0,\mathbf{x}_2,0)\phi(0,\mathbf{x}_2,0)}{1-\gamma_1}}{\gamma_3 \phi(1, \mathbf{x}_2, 0)} \tag{34}$$

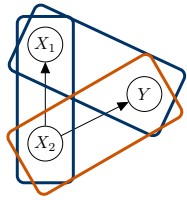

Figure 5: An example of the scenarios considered in Theorem 2

Note we have the adjusted coefficients are consistent with each other:

$$(1 - \gamma_1)c(1, \mathbf{x}_2, 0)\phi(1, \mathbf{x}_2, 0) = \frac{1}{\gamma_3}\big[(1 - \gamma_1)f_s(1, \mathbf{x}_2) - (1 - \gamma_3)f_t(\mathbf{x}_2, 1) \tag{35}$$

$$+ \gamma_1 f_s(0, \mathbf{x}_2) - \gamma_1\gamma_3 c(0, \mathbf{x}_2, 0)\phi(0, \mathbf{x}_2, 0)\big] \tag{36}$$

$$= f_t(\mathbf{x}_2, 0) - \gamma_1 c(0, \mathbf{x}_2, 0)\phi(0, \mathbf{x}_2, 0) \tag{37}$$

Thus it is possible to find a new $\phi'$ such that it still satisfies the system of equations. More over $\phi'$ deviates from $\phi$ arbitrarily large by the choice of $c(0, \mathbf{x}_2, 0)$:

$$||\phi - \phi'||_2 \geq ||\phi(0, \mathbf{x}_2, 0) - c(0, \mathbf{x}_2, 0)\phi(0, \mathbf{x}_2, 0)||_2 \geq ||c - 1||_2 * \text{const} \tag{38}$$

□

## C.3  THEOREM 2

### C.3.1  SINGLE-VARIATE STATEMENT

**Theorem 6.** *Consider a target variable $Y$ and its direct cause $PA_Y$. Suppose that we observe:*

- *source environment contains variables $(Z, Y)$; training a discriminative model on this environment yields a function $f_s(z) = \mathbb{E}[Y \mid Z]$,*

- *target environment contains variable $PA_Y$*

*Suppose $Y := \phi(PA_Y) + \epsilon_Y$, $Z = g(PA_Y) + \epsilon_Z$ where $g$ is known and invertible with $\phi, g^{-1}$ uniformly continuous. Then in the limit of $\mathbb{E}[|\epsilon_Z|] \to 0$, the composition of the discriminative models in source environments also approaches the optimal predictor, i.e., $\forall pa_Y : f_s \circ g(pa_Y) \to \phi(pa_Y)$.*

### C.3.2  MULTIVARIATE STATEMENT

**Theorem 6.** *Consider a target variable $Y$ and its direct causes $\mathbf{PA}_Y$. Suppose that we observe:*

- *source environment contains variables $(\mathbf{X}_s, Y)$; training a discriminative model on this environment yields $f_s(\mathbf{x}_s) = \mathbb{E}[Y \mid \mathbf{X}_s]$,*

- *target environment contains variables $\mathbf{X}_t = \mathbf{PA}_Y$*

*Suppose $Y := \phi(\mathbf{PA}_Y) + \epsilon_Y$ and $\mathbf{X}_s = g(\mathbf{X}_t) + \epsilon_s$ where $g$ is known and invertible with $\phi, g^{-1}$ are uniformly continuous and $\epsilon_s \perp\!\!\!\perp \mathbf{X}_t$. Then in the limit of $\mathbb{E}[|\epsilon_s|] \to 0$, the composition of $f_s \circ g$ approaches the optimal predictor, i.e., $\forall \mathbf{x}_t : f_s \circ g(\mathbf{x}_t) \to \phi(\mathbf{x}_t)$.*

### C.3.3  PROOF

*Proof.* We first observe the optimal predictive function in the target environment coincides with the true generating function, written as: $f_t(\mathbf{PA}_Y) = \mathbb{E}[Y \mid \mathbf{PA}_Y] = \phi(\mathbf{PA}_Y)$. Further, $g(\mathbf{X}_t) = \mathbb{E}[\mathbf{X}_s \mid \mathbf{X}_t]$ due to $\epsilon_s \perp\!\!\!\perp \mathbf{X}_t$. Given $g$ is continuous and invertible, its inverse $g^{-1}$ exists and is continuous.

$$\mathbb{E}[Y \mid \mathbf{x}_s] = \mathbb{E}\big[\mathbb{E}[Y \mid \mathbf{x}_s, \mathbf{X}_t]\big] \tag{39}$$

$$= \mathbb{E}_{\mathbf{PA}_Y \mid \mathbf{x}_s}\big[\mathbb{E}[Y \mid \mathbf{PA}_Y]\big] \tag{40}$$

$$= \mathbb{E}_{\mathbf{PA}_Y \mid \mathbf{x}_s}\big[\phi(\mathbf{PA}_Y)\big] \tag{41}$$

Assume $\phi, g^{-1}$ is uniformly continuous. Then $\phi g^{-1}$ is uniformly continuous, i.e., for any $\delta \in \mathbb{R}$, there exists $\gamma$, such that for any $x \in \mathbb{R}$,

$$|\phi g^{-1}(x + \delta) - \phi g^{-1}(x)| \le \gamma|\delta|$$

Then,

$$\mathbb{E}_{\mathbf{PA}_Y \mid \mathbf{x}_s}\big[\phi(\mathbf{PA}_Y)\big] = \mathbb{E}_{\mathbf{PA}_Y \mid \mathbf{x}_s}\big[\phi g^{-1}(\mathbf{x}_s - \epsilon_s)\big] \tag{42}$$

By the uniform continuity of $\phi g^{-1}$, $|\mathbb{E}[Y \mid \mathbf{x}_s] - \phi g^{-1}(\mathbf{x}_s)| = |\mathbb{E}_{\mathbf{PA}_Y \mid \mathbf{x}_s}\big[\phi g^{-1}(\mathbf{x}_s - \epsilon_s) - \phi g^{-1}(\mathbf{x}_s)\big]| \le \mathbb{E}_{\mathbf{PA}_Y \mid \mathbf{x}_s}\big[|\phi g^{-1}(\mathbf{x}_s - \epsilon_s) - \phi g^{-1}(\mathbf{x}_s)|\big] \le \gamma \mathbb{E}[|\epsilon_s|]$. In the limit of $\mathbb{E}[|\epsilon_s|] \to 0$, the result follows. $\qquad \square$

### C.4 THEOREM 3

#### C.4.1 SINGLE-VARIATE STATEMENT

**Theorem 3.** *Consider the problem setup in § 3.1 and assume the function $\phi$ is everywhere twice differentiable with respect to $X_3$. Suppose from the source environment we learn a function $f_s(x_1, x_2) = \mathbb{E}[Y \mid x_1, x_2]$. Using first-order Taylor approximation on the function $\phi : x_1 \times x_2 \times \mathcal{X}_3 \to \mathbb{R}$ for fixed $x_1, x_2$, the moments of the residual distribution in the source environment take the form*

$$\mathbb{E}[(Y - f_s(x_1, x_2))^n \mid x_1, x_2] = \sum_{k=0}^{n} \binom{n}{k} \mathbb{E}[\epsilon^k] \left(\frac{\partial \phi}{\partial X_3}\bigg|_{x_1, x_2, \mu_3}\right)^{n-k} \mathbb{E}[(X_3 - \mu_3)^{n-k}]. \tag{11}$$

*For $n = 3$, this reduces to*

$$\mathbb{E}[(Y - f_s(x_1, x_2))^3 \mid x_1, x_2] = \left(\frac{\partial \phi}{\partial X_3}\bigg|_{x_1, x_2, \mu_3}\right)^3 \mathbb{E}[(X_3 - \mu_3)^3] + \mathbb{E}[\epsilon^3]. \tag{12}$$

#### C.4.2 MULTIVARIATE STATEMENT

**Theorem 7.** *Consider the problem setup in § 3.1, and assume the function $\phi$ be a 2-times continuously differentiable function at the point $\mu_{\mathbf{PA}_Y} := \mathbb{E}[\mathbf{PA}_Y]$. Suppose from the source environment we learn a function $f_s(\mathbf{x}_s) = \mathbb{E}[Y \mid \mathbf{x}_s]$. Denote the $r$-th central moment of $X_i$ as $C_i^r = \mathbb{E}[(X_i - \mu_i)^r]$. Using first-order multivariate Taylor approximation on the function $\phi : \mathbf{x}_s \times \mathbf{X}_{t \setminus o} \to \mathbb{R}$, denoted as $\phi\big|_{\mathbf{x}_s}$ and suppose $\mathbf{X}_{t \setminus o}$ have dimension $m$, the moments of the residual distribution in the source environment take the form, where $f = \phi\big|_{\mathbf{x}_s}$ for ease of notation,*

$$\mathbb{E}[(Y - f_s(\mathbf{x}_s))^n \mid \mathbf{x}_s] = \sum_{k=0}^{n} \binom{n}{k} \mathbb{E}[\epsilon_Y^k] \times \Bigg[ \sum_{k_1 + k_2 + \cdots + k_m = n-k; k_1, k_2, \ldots, k_m \ge 0} \tag{43}$$

$$\binom{n}{k_1, k_2, \ldots, k_m} \prod_{i=1}^{m} (\frac{\partial f}{\partial x_i}(\mathbf{a}))^{k_i} C_i^{k_i} \Bigg], \quad \textit{where } \mathbf{a} = \mu_{\mathbf{x}_{t \setminus o}} \tag{44}$$

*When $n = 3$:*

$$\mathbb{E}[(Y - f_s(\mathbf{x}_s))^3 \mid \mathbf{x}_s] = \sum_{i=1}^{m} (\frac{\partial f}{\partial x_i}(\mathbf{a}))^3 C_i^3 + \mathbb{E}[\epsilon_Y^3] \tag{45}$$

### C.4.3 PROOF

**Notation** Let $|\alpha| = \sum_i \alpha_i, \alpha! = \prod_i \alpha_i!, \mathbf{x}^\alpha = \prod_i x_i^{\alpha_i}$ for $\alpha \in \mathbb{N}^n$ and $\mathbf{x} \in \mathbb{R}^n$. Denote

$$D^\alpha f = \frac{\partial^{|\alpha|} f}{\partial x_1^{\alpha_1} \dots \partial x_n^{\alpha_n}}$$

as higher order partial derivatives of $f$.

**Theorem 8** (Multivariate version of Taylor's theorem (Spivak, 2008)). *Let $f : \mathbb{R}^n \to \mathbb{R}$ be a k-times continuously differentiable function at the point $\mathbf{a} \in \mathbb{R}^n$. Then there exist functions $h_\alpha : \mathbb{R}^n \to \mathbb{R}$, where $|\alpha| = k$, such that*

$$f(\mathbf{x}) = \sum_{|\alpha| \le k} \frac{D^\alpha f(\mathbf{a})}{\alpha!} (\mathbf{x} - \mathbf{a})^\alpha + \sum_{|\alpha| = k} h_\alpha(\mathbf{x})(\mathbf{x} - \mathbf{a})^\alpha, \tag{46}$$

$$and \quad \lim_{\mathbf{x} \to \mathbf{a}} h_\alpha(\mathbf{x}) = 0 \tag{47}$$

*Proof.* Let $\mathbf{x}$ be a sample from random variable $\mathbf{X}$. Let $\mathbf{a} = \mathbb{E}[\mathbf{X}]$. Theorem 8 states that

$$f(\mathbf{x}) = f(\mathbf{a}) + \sum_{|\alpha|=1} Df(\mathbf{a})(\mathbf{x} - \mathbf{a}) + \sum_{|\alpha|=2} h_\alpha(\mathbf{x})(\mathbf{x} - \mathbf{a})^2$$

Take first-order Taylor approximation over the generating function, we suppose $f(\mathbf{x}) \approx f(\mathbf{a}) + Df(\mathbf{a})(\mathbf{x} - \mathbf{a}), \forall \mathbf{x}$. Taking expectations:

$$\mathbb{E}[f(\mathbf{X})] \approx f(\mathbf{a}) \tag{48}$$

Consider the difference between $f(x)$ and its expectations and raise it to the power of $n$, we have:

$$\left( f(\mathbf{X}) - \mathbb{E}[f(\mathbf{X})] \right)^n = \left( \sum_{|\alpha|=1} D^\alpha f(\mathbf{a})(\mathbf{X} - \mathbf{a})^\alpha \right)^n \tag{49}$$

$$= \left( \sum_{i=1}^m \frac{\partial f}{\partial x_i}(\mathbf{a})(X_i - a_i) \right)^n \tag{50}$$

Taking expectations, on Eq. 49, and let $\mathbf{X} \in \mathbb{R}^m$, we have:

$$\mathbb{E}\left[ \left( f(\mathbf{X}) - \mathbb{E}[f(\mathbf{X})] \right)^n \right] = \sum_{k_1+k_2+\dots+k_m=n; k_1,k_2,\dots,k_m \ge 0} \binom{n}{k_1, k_2, \dots, k_m} \prod_{i=1}^m (\frac{\partial f}{\partial x_i}(\mathbf{a}))^{k_i} \mathbb{E}\left[ (X_i - a_i)^{k_i} \right] \tag{51}$$

The expectation is taken inside the product term as the covariates are independent of each other. Take $f_{\mathbf{x}_s} : \mathbf{X}_{t \setminus o} \to \mathbb{R}$ to be the function $\phi : \mathbf{x}_s \times \mathbf{X}_{t \setminus o} \to \mathbb{R}$ where values $\mathbf{x}_s$ are fixed. Since $Y = \phi(\mathbf{x}_s, \mathbf{x}_{t \setminus o}) + \epsilon$, we have

$$\mathbb{E}[(Y - f_s(\mathbf{x}_s))^n \mid \mathbf{x}_s] = \mathbb{E}\left[ \left( f(\mathbf{x}) + \epsilon - \mathbb{E}[f(\mathbf{x})] \right)^n \right] \tag{52}$$

$$= \sum_{k=0}^n \binom{n}{k} \mathbb{E}[\epsilon^k] \mathbb{E}[(f(\mathbf{x}) - \mathbb{E}[f(\mathbf{x})])^{n-k}] \tag{53}$$

where $f(\mathbf{x}) = f_{\mathbf{x}_s}(\mathbf{x}_{t \setminus o})$. The second equality is due to independence of $\epsilon$ and $f(x) - \mathbb{E}[f(x)]$. Let $C_i^r$ denotes the $r$-th central moment of $X_i$ where $C_i^r := \mathbb{E}[(X_i - \mu_i)^r]$. Substitute in Eq. 51, we have

$$\mathbb{E}[(Y - f_s(\mathbf{x}_s))^n \mid \mathbf{x}_s] = \sum_{k=0}^n \binom{n}{k} \mathbb{E}[\epsilon_Y^k] \times \Bigg[ \sum_{k_1+k_2+\dots+k_m=n-k; k_1,k_2,\dots,k_m \ge 0} \tag{54}$$

$$\binom{n}{k_1, k_2, \dots, k_m} \prod_{i=1}^m (\frac{\partial f}{\partial x_i}(\mathbf{a}))^{k_i} C_i^{k_i} \Bigg] \tag{55}$$

When $n = 3$:

$$\mathbb{E}[(Y - f_s(\mathbf{x}_s))^3 \mid \mathbf{x}_s] = \sum_{i=1}^m (\frac{\partial f}{\partial x_i}(\mathbf{a}))^3 C_i^3 + \mathbb{E}[\epsilon_Y^3] \tag{56}$$

$\square$

## C.5 COROLLARY 4

**Corollary 9.** *For OOV scenarios described in § 3.1, learning from the moment of the error distribution allows exact identification of $\phi$ when $\phi(x_1, x_2) = \sum_{p,q} c_i h(x_1, x_2)^p x_3^q$, where $p, q \in \{0, 1\}$ and $c_i \in \mathbb{R}, \forall i$ and $h$ can be any function.*

### C.5.1 PROOF

*Proof.* When $\phi(x_1, x_2) = \sum_{p,q} c_i h(x_1, x_2)^p x_3^q$, where $p, q \in \{0, 1\}$ and $c_i \in \mathbb{R}, \forall i$ and $h$ can be any function. Note

$$f_s(x_1, x_2) = c_1 + c_2 x_3 + c_3 h(x_1, x_2) + c_4 h(x_1, x_2) x_3$$

By Theorem 7, when $m = 1, n = 3$

$$\mathbb{E}[(Y - f_s(x_1, x_2))^3 \mid x_1, x_2] = \left(\frac{\partial \phi}{\partial x_3}\big|_{x_1, x_2, \mu_3}\right)^3 C_3^3 + \mathbb{E}[\epsilon_Y^3] \tag{57}$$

$$\frac{\partial \phi}{\partial x_3}\big|_{x_1, x_2, \mu_3} = c_2 + c_4 h(x_1, x_2) \tag{58}$$

Then $\phi(x_1, x_2, x_3) = f_s(x_1, x_2) + \frac{\partial \phi}{\partial x_3}\big|_{x_1, x_2, \mu_3} * (x_3 - \mu_3)$, where $f_s$ estimable from the source environment, and the partial derivative estimable from the residual error distribution and $\mu_3$ estimable from the covariates in the target environment.

□

## C.6 EXTENSIONS TO MORE THAN ONE UNOBSERVED VARIABLE

Here, we consider a function with two variables $f(x, y)$ where variables can be considered as two unobserved variables from the source environment. Note, the same argument can easily extend to multivariate functions. Let $\mathbb{E}[X] = \mu_x, \mathbb{E}[Y] = \mu_y$. Expand multivariate Taylor approximations around the point $\mathbf{a} = (\mu_x, \mu_y)$, we have:

$$f(x, y) = f(\mu_x, \mu_y) + \frac{\partial f}{\partial x}\big|_{\mathbf{a}}(x - \mu_x) + \frac{\partial f}{\partial y}\big|_{\mathbf{a}}(y - \mu_y) \tag{59}$$

$$+ C_1(x - \mu_x)^2 + C_2(x - \mu_x)(y - \mu_y) + C_3(y - \mu_y)^3 \tag{60}$$

With first-order Taylor approximations, we ignore the higher order terms. Taking expectations on both sides, we have $\mathbb{E}[f(x, y)] = f(\mu_x, \mu_y)$. Similarly,

$$\left(f(x, y) - \mathbb{E}[f(x, y)]\right)^n = \left(\frac{\partial f}{\partial x}\big|_{\mathbf{a}}(x - \mu_x) + \frac{\partial f}{\partial y}\big|_{\mathbf{a}}(y - \mu_y)\right)^n \tag{61}$$

$$= \sum_{k=0}^{n} \binom{n}{k} \left(\frac{\partial f}{\partial x}\big|_{\mathbf{a}}\right)^k (x - \mu_x)^k \left(\frac{\partial f}{\partial y}\big|_{\mathbf{a}}\right)^{n-k} (y - \mu_y)^{n-k} \tag{62}$$

Taking expectations on both sides, assuming we can estimate the cross-moments between two unobserved variables from data, with two unknowns and two equations, we can estimate the unknowns.

## D FURTHER EXPERIMENTAL DETAILS

### D.1 IMPLEMENTATION DETAILS

In the implementation of the mean imputed predictor, we first impute the missing variable $X_3$ with its mean and train a source predictor $f_s$ from $X_1, X_2, \mathbb{E}[X_3]$. During inference, given a target sample $(x_2, x_3)$, MeanImputed$(x_2, x_3) = f_s(\mu_1, x_2, x_3)$ where $\mu_1 := \mathbb{E}[X_1]$.

In the implementation of the marginal predictor, we first train a source predictor $f_s$ with inputs $X_1, X_2$. During inference, given target sample $(x_2, x_3)$, Marginal$(x_2, x_3) = \sum_{x_{1,i}} f_s(x_{1,i}, x_2)$.

For all our training, we employ a 2-layer MLP with ReLU activation function. All MLPs are trained to minimize mean squared error loss using SGD. For the Monte Carlo approximation in our proposed *MomentLearn* we sample $1,000$ observations of $X_1$ from the source environment.

## D.2 HYPERPARAMETER SWEEP

We have performed a hyperparameter sweep for a total of 8 variations where learning rate varies in $(0.01, 0.001)$, hidden sizes in the range of $(64, 32)$ and the number of epochs in the range of $(30, 50)$. Table 1 shows the averaged results over the 8 variations.

## D.3 ALGORITHM

Below we detail the exact algorithm for performing OOV learning in our base model illustrated in Fig. 1a. Note that, we train two neural networks: one to estimate the conditional mean in the source environment $f_s$, and the other to estimate the partial derivative from modelling the third moment of the residual distributions. We use 2-layer MLPs with ReLU activation function with hidden size 64 and output size 1. We train with batch size 64, learning rate 0.01 with weight decay $1e^{-4}$. We train the conditional mean estimator for 10 epochs and the partial derivative estimator for 50 epochs. We perform Monte Carlo estimation using 1000 samples. We sample our data ensuring that the coefficients for the missing variable are large enough, i.e., $|\alpha_3| > 2(|\alpha_2| + |\alpha_1|)$ for performance analysis and sample efficiency experiment. Otherwise, we sample coefficients from a standard normal distribution with mean 0 and variance 1 for the systematic analysis experiment.

---

**Algorithm 1:** Out-of-variable learning

**Input** : Source environment $\mathcal{E}_S$ with variables $X_1$, $X_2$ and $Y$; Target environment $\mathcal{E}_t$ with variables $X_2$ and $X_3$.
**Output** : OOV predictive function $\tilde{f}_t(x_2, x_3)$

1 **Step 1**: Learn $\mathbb{E}[Y \mid X_1, X_2]$
2 Train a neural network $f_s$ via minimizing its mean squared error $||Y - f_s(x_1, x_2)||_2^2$
3 **Step 2**: Learn partial derivative $h_\theta$ from modelling conditional skew
4 Compute $Z = (Y - f_s(X_1, X_2))^3$.
5 Estimate the skew of $X_3$: $k_3 = \mathbb{E}[(X_3 - \mu_3)^3]$, where $\mu_3 = \mathbb{E}[X_3]$.
6 Train a neural network $h_\theta$ via minimizing $||Z - k_3 h_\theta(x_1, x_2)^3||_2^2$
7 **Step 3**: Monte Carlo Estimation
8 Uniformly sample $n$ observations of $X_1$ from environment $\mathcal{E}_S$: $\{x_{1,i}\}_{i=1}^n$.
9 For fixed $x_2, x_3$, calculate the proposed zero-shot estimate in Eq. 14.

---

## D.4 REAL WORLD EXPERIMENT

To illustrate the applicability of OOV generalization in real world dataset, we use "mtcars" dataset extracted from 1974 Motor Trend US magazine. Given the small dataset size, we first augmented the dataset through resampling with replacement to reach 232 data points. We are interested in predicting the outcome variable $Y$ miles per gallon (MPG) given variables on the car's information. We split the source and target dataset with 80-20 ratio. In the source environment, we observed the number of cylinders $X_1$ and quarter-mile time (acceleration) $X_2$ and miles per gallon $Y$. In the target environment, we observe covariates quarter-mile time $X_2$ and weight of the car $X_3$. We are interested in leveraging observation from the source environment to yield a better prediction on the target covariates without observation of the outcome in the target environment. Averaged over 10 random seeds, Table 2 shows the zero-shot prediction for our method and various benchmarks.

|  | Mtcars |
| --- | --- |
| MomentLearn | $1.09 \pm 0.08$ |
| MeanImputed | $1.48 \pm 0.06$ |
| Marginal | $1.46 \pm 0.03$ |

Table 2: Our method's ("MomentLearn") OOV prediction performance in the target environment, compared to the "Marginal" baseline and the predictor that imputes missing variable with its mean ("Mean Imputed"). Shown are mean and variance of the MSE loss between the predicted and observed target values on augmented 'Mtcars' dataset.

### D.5 ROBUSTNESS WITH DIFFERENT NOISE SCALE

To understand the robustness of our method with changing noise level, we vary the standard deviation of Gaussian noise (with mean 0) in the range of $\sigma = [0.01, 0.2, 0.4, 0.6, 0.8, 1.0]$. For each noise setting, we repeat the experiment for 5 random seeds and take the average of MSE loss for each predictor. Shown are mean and standard deviations of the MSE loss between the predicted and observed target values in Table 3. We observe MomentLearn outperforms the other baselines for almost all cases.

| | $\sum_i \alpha_i X_i$ | | | | | |
|---|---|---|---|---|---|---|
| | $\sigma = 0.01$ | $\sigma = 0.2$ | $\sigma = 0.4$ | $\sigma = 0.6$ | $\sigma = 0.8$ | $\sigma = 1.0$ |
| Oracle | $0.30 \pm 0.15$ | $0.34 \pm 0.14$ | $0.46 \pm 0.14$ | $0.67 \pm 0.14$ | $0.96 \pm 0.13$ | $1.33 \pm 0.13$ |
| MomentLearn | $0.31 \pm 0.15$ | $0.35 \pm 0.15$ | $0.51 \pm 0.19$ | $0.80 \pm 0.18$ | $1.13 \pm 0.18$ | $1.52 \pm 0.17$ |
| MeanImputed | $0.44 \pm 0.20$ | $0.48 \pm 0.20$ | $0.61 \pm 0.18$ | $0.81 \pm 0.17$ | $1.11 \pm 0.15$ | $1.48 \pm 0.13$ |
| Marginal | $0.54 \pm 0.28$ | $0.58 \pm 0.27$ | $0.91 \pm 0.24$ | $0.88 \pm 0.45$ | $1.20 \pm 0.22$ | $1.57 \pm 0.20$ |
| | $+ \sum_{i<j} \beta_{ij} X_i X_j$ | | | | | |
| | $\sigma = 0.01$ | $\sigma = 0.2$ | $\sigma = 0.4$ | $\sigma = 0.6$ | $\sigma = 0.8$ | $\sigma = 1.0$ |
| Oracle | $0.25 \pm 0.45$ | $0.29 \pm 0.45$ | $0.43 \pm 0.44$ | $0.64 \pm 0.43$ | $0.93 \pm 0.42$ | $1.30 \pm 0.40$ |
| MomentLearn | $0.29 \pm 0.43$ | $0.31 \pm 0.44$ | $0.45 \pm 0.43$ | $0.66 \pm 0.42$ | $0.96 \pm 0.41$ | $1.33 \pm 0.39$ |
| MeanImputed | $0.49 \pm 0.49$ | $0.54 \pm 0.49$ | $0.66 \pm 0.45$ | $0.88 \pm 0.47$ | $1.18 \pm 0.46$ | $1.57 \pm 0.45$ |
| Marginal | $0.51 \pm 0.45$ | $0.56 \pm 0.45$ | $0.68 \pm 0.45$ | $0.90 \pm 0.44$ | $1.58 \pm 0.42$ | $1.46 \pm 0.19$ |
| | $+ \sum_i \gamma_i X_i^2$ | | | | | |
| | $\sigma = 0.01$ | $\sigma = 0.2$ | $\sigma = 0.4$ | $\sigma = 0.6$ | $\sigma = 0.8$ | $\sigma = 1.0$ |
| Oracle | $0.56 \pm 0.32$ | $0.60 \pm 0.32$ | $0.72 \pm 0.31$ | $0.92 \pm 0.31$ | $1.21 \pm 0.31$ | $1.59 \pm 0.31$ |
| MomentLearn | $0.71 \pm 0.33$ | $0.94 \pm 0.45$ | $0.81 \pm 0.29$ | $0.99 \pm 0.29$ | $1.32 \pm 0.24$ | $1.68 \pm 0.22$ |
| MeanImputed | $1.37 \pm 0.95$ | $1.43 \pm 0.95$ | $1.56 \pm 0.96$ | $1.77 \pm 0.95$ | $2.08 \pm 0.97$ | $2.46 \pm 0.98$ |
| Marginal | $1.74 \pm 1.27$ | $1.78 \pm 1.27$ | $1.91 \pm 1.28$ | $2.13 \pm 1.28$ | $2.42 \pm 1.29$ | $2.80 \pm 1.30$ |

Table 3: Under changing noise level where noise sampled from Gaussian distribution with varying standard deviation $\sigma$, our method's ("MomentLearn") OOV prediction performance in the target environment, compared to the "Marginal" baseline, the predictor that imputes missing variable with its mean ("Mean Imputed") and the solution that has access to the full joint observations on the target domain ("Oracle"). Shown are mean and standard deviations of the MSE loss between the predicted and observed target values.

### D.6 ROBUSTNESS WITH HEAVY TAILED

To understand the robustness of our method with non-Gaussian noise, we sample noise from lognormal distribution with mean 0 and $\sigma = 0.5$ and repeat the experiment for 5 times averaged over a hyperparameter sweep. We see a decrease in performance for our method as expected by Theorem 3 due to the entanglment of noise skew with the signal skew. Table 4 shows the detailed result.

## E DISCUSSION

### E.1 MORE ENVIRONMENTS

To understand how multi-environments could in some cases help the OOV problem, recall Theorem 2 where the dependence structure among covariates is assumed to be known. Such an assumption can be replaced with a realistic scenario where we observe all input variables for the source and

|  | $\sum_i \alpha_i X_i$ | $+\sum_{i<j} \beta_{ij} X_i X_j$ | $+\sum_i \gamma_i X_i^2$ |
|---|---|---|---|
| Oracle | $0.69 \pm 0.14$ | $0.63 \pm 0.43$ | $0.93 \pm 0.33$ |
| MomentLearn | $0.91 \pm 0.34$ | $0.97 \pm 0.50$ | $1.20 \pm 0.30$ |
| MeanImputed | $0.89 \pm 0.31$ | $1.00 \pm 0.53$ | $1.76 \pm 0.86$ |
| Marginal | $0.93 \pm 0.23$ | $0.90 \pm 0.45$ | $2.11 \pm 1.30$ |
|  | $\mathcal{GP}_1(X_1, X_2) + \alpha_3 X_3$ | $+\mathcal{GP}_2(X_1, X_2) \cdot X_3$ | $+\mathcal{GP}_3(X_1, X_2) \cdot X_3^2$ |
| Oracle | $0.39 \pm 0.03$ | $0.44 \pm 0.08$ | $0.54 \pm 0.16$ |
| MomentLearn | $0.82 \pm 0.35$ | $0.72 \pm 0.21$ | $1.40 \pm 0.66$ |
| MeanImputed | $0.85 \pm 0.42$ | $0.78 \pm 0.41$ | $1.31 \pm 0.95$ |
| Marginal | $0.65 \pm 0.15$ | $0.79 \pm 0.39$ | $1.81 \pm 1.22$ |

Table 4: Under heavy tailed noise sampled from lognormal distribution with $\mu = 0$ and $\sigma = 0.5$, our method's ("MomentLearn") OOV prediction performance in the target environment, compared to the "Marginal" baseline, the predictor that imputes missing variable with its mean ("Mean Imputed") and the solution that has access to the full joint observations on the target domain ("Oracle"). Shown are mean and standard deviations of the MSE loss between the predicted and observed target values. $\mathcal{GP}_i(\cdot)$ denotes a function sampled from a Gaussian Process with zero mean and Gaussian kernel. MomentLearn performs as expected by our theoretical results and even exhibits a degree of robustness to function classes that are not covered by Theorem 3.

target environments in another environment and thus estimate $g$ through learning in this environment. If additional environments contain covariates unique to the target environment and covariates in the source environment, such information is in general helpful. One can thus learn their functional relationship and impute with the estimated value to achieve a more accurate predictor in the target environment.

## E.2 Assumptions

To facilitate a full understanding of our theorems, we provide a bullet list of assumptions required and discuss their implications and robustness to their violations. Here we focus on uni-variate discussion and multivariate extension is easy to generalize.

Theorem 3 presents an analytical formula on how the moments of the residual distribution relate to transferable signals (partial derivative in Eq. 10), moments of the out-of-variable, and noise effect. Assumptions involved are:

- continuous covariates $\mathbf{X}$ are causes of the outcome variable $Y$ and $Y = \phi(\mathbf{X}) + \epsilon$
- $\phi$ is everywhere twice-differentiable with respect to the out-of-variable $X_3$

Corollary 4 presents an identification result on when our method "MomentLearn" can achieve perfect transferring ability. Assumptions involved are:

- continuous covariates $\mathbf{X}$ are causes of the outcome variable $Y$ and $Y = \phi(\mathbf{X}) + \epsilon$
- $\phi$ satisfies $\{\phi | \phi(\mathbf{x}) = \sum_{p,q} c_{p,q} h(x_1, x_2)^p x_3^q, p, q \in \{0, 1\}, c_i \in \mathbb{R}, \forall h\}$
- noise $\epsilon$ is symmetric

### E.2.1 Robustness to violations of assumptions

**Causal assumptions** We study the OOV problem under a causal framework. As discussed in Section § 2, while there is nothing causal about the OOV problem, we utilize structural causal model to study cases that provably exhibit OOV generalization. If no knowledge about the graph is available, then things can go arbitrarily wrong. E.g., the relationship of $X_3$ can be arbitrarily related to the target variable and this cannot be inferred from the source environment unless further assumptions are made (as indicated in Section 3.3.1). It is conceivable that results could be obtained in broader settings,

e.g., if the target covariates have relationships with the source covariates in a more complex causal graph, partial information may be recoverable, though it is out of scope for the current paper.

**Robustness to function class** We performed systematic analysis on how our method performs with respect to different function classes in Section 4 with results record in Table 1. We observe "MomentLearn" performs as expected by our theoretical results and even exhibits a degree of robustness to function classes that are not guaranteed as in Corrolary 4.

**Robustness to noise** We perform further experimental analysis on how our method performs when the standard deviation of Gaussian noise increases and when the noise is asymmetric (e.g., follows a log-normal distribution). Table 3 and 4 records the result. We observe "MomentLearn" performs as expected by our theoretical results: consistently outperforms other baselines facing Gaussian noise with increasing noise levels but deteriorates in performance when noise is asymmetric.

### E.3 OOD VS. OOV AND ITS APPLICATIONS

Here we provide a brief discussion on OOD and OOV's relationship and ground the OOV problem in potential real-world applications.

Under no distribution shift, problems can exhibit the need to generalize OOV. This is evident in real-world scenarios as datasets are often inconsistent. For example, consider two medical labs collecting different sets of variables. Lab A collects $X_1$ = lifestyle factors and $X_2$ = blood test; Lab B, in addition to $X_2$, collects $X_3$ = genomics. Lab A is hospital-based with data capacity, whereas lab B is research-focused. The OOV problem asks: given a model trained to predict the likelihood of a disease $Y$ on Lab A's data, how should Lab B use this model for its own dataset that differs in the set of input variables? Situations as described often happen in the real-world (e.g. hospitals, consumer industries) as different institutions have imbalanced resources to collect data and have a different market focus which reflects on the type of variables collected.

Problems exhibit distribution shifts may also be due to hidden OOV problems. Guo et al. (2022) provides theoretical evidence that exchangeable sequences of causal observations (i.e., a set of causal observations that come from different distributions and satisfy exchangeability) can be equivalently modelled as a set of identical distributions conditioned on latent variables. One may thus interpret distribution shifts as a lack of knowledge of the latent variable. In practice, for example, different treatment effects on patients may be due to unobserved variables idiosyncratic to individual patients.

Often in real-world applications, problems exhibit both OOD and OOV. For example, to assess the effect of a policy, decision-makers need to synthesize information from multiple sources containing different variables, and account remaining randomness as distribution shifts for risk measure. To effectively tackle real world problems, with the power of AI, we believe one need to solve both OOD and OOV problems. We envision this work is conceptually novel, explicating the capability of generalization is intricately related to the knowledge of variables and their relationships. We think this is likely to trigger significant follow-up work.