# OpenReview forum: "Out-of-Variable Generalisation for Discriminative Models"
_ICLR.cc/2024/Conference — ICLR 2024 poster_

### Official Review · Reviewer_qFZD · 2023-10-31

**Soundness:** 3 good
**Presentation:** 3 good
**Contribution:** 2 fair
**Rating:** 5
**Confidence:** 2

**Summary:**

The paper introduces out-of-variable (OOV) generalization, which is an the ability to generalize in environments with variables that have never been jointly observed before. OOV is an issue in settings where different variables (e.g. diagnostic tests) are available for different environments (e.g. different patients). The paper investigates challenges for common approaches when faced with the OOV problem, and proposes an OOV predictor that leverage moments of the error distribution. The work contributes to theoretical understandings of OOV and offers a proof-of-concept for a predictor capable of non-trivial OOV transfer.

**Strengths:**

- The paper formally studies a new perspective on generalization.
- The methods employed in the paper are sound.

**Weaknesses:**

- The paper does not demonstrate the practical applicability of the concept of OOV generalization, and the setting feels a bit contrived. Also it seems like OOV generalization can be thought of just a case of OOD generalization--if we think about all the variables together as the input, the OOV generalization is just a case of OOD generalization (e.g. covariate shift) where some inputs have clear signal from some features and other inputs have clear signal from other features.
- It would be helpful to include more intuitive discussion throughout the paper providing more analysis on the sections. For example, more discussion on the assumptions of the settings/theorems would be helpful, and it's not clear exactly under what assumptions the proposed predictor is appropriate.

**Questions:**

Please see weaknesses above.

---

> ### Author Response · Authors · 2023-11-14
>
> > Also it seems like OOV generalization can be thought of just a case of OOD generalization--if we think about all the variables together as the input, the OOV generalization is just a case of OOD generalization (e.g. covariate shift) where some inputs have clear signal from some features and other inputs have clear signal from other features.
>
> We are sorry for the misunderstanding. We would like to clarify OOV and OOD are quite distinct generalization notions. Covariate shift, for example, considers the input space $X$ remains the same but the distribution on input space shifts from source to target $P_s(X)$ to $P_t(X)$. However, OOV considers the input space between source and target contains different sets of variables, even though they could come from the same data generating process, as illustrated in Fig. 1a where X_s = [X1, X2] and X_t = [X2, X3], but X1, X2, X3 all come from the same distribution P(X1, X2, X3).  As a concrete example, each country records own population’s vaccine data, e.g., [Age, Is_vaccined]. Such dataset have different distributions for each country (OOD) but share the same set of variables (In variable). However, if hospital A in the same country records their patients vaccine data as [Age, city, is_vaccined], and hospital B in the same country records data as [Age, health_history, is_vaccined], then though they share the same distribution as coming from the same country (in-distribution) they have different variable sets (OOV). We refer to Appendix E.3 for a more detailed discussion on OOD and OOV.
>
> > The paper does not demonstrate the practical applicability of the concept of OOV generalization
>
> We thank you for your comment, though we hope to demonstrate the practical applicability of OOV through examples in AI in medicine (re introduction) and provide a concrete example in section 3.1. As the other reviewers also acknowledges our efforts in explicitly grounding the concept, for example, reviewer xwQY acknowledges that the paper provides “simple, clear real-world examples to elucidate the problem”, reviewer woi7 comments that “examples also made things concrete and easy to follow” and reviewer s9Ga thinks we “provide an extensive study … including…examples”.
>
> We think your comment is maybe more directed on the lack of real world dataset. We carefully take your feedback into consideration and following your suggestion, we include an additional real world experiment with details in Appendix D.4.
>
> > the setting feels a bit contrived.
>
> We are sorry for this impression, though this paper, unlike many mainstream papers, aims to navigate a new research direction and inspire the community on a surprising overlooked but important research area.
>
> We studied how common transfer learning approaches would fail in OOV setting and show under what circumstance identifiability is achievable for dependent and independent covariates. We exposes a problem that, in the stated form, may appear unsolvable at first sight, and then goes on to provide a mathematically valid and experimentally corroborated solution. In addition, we openly discuss our limitations and point the readers to potential avenues of future research for the area (re conclusion).
>
> We feel that the paper is already useful to the community at this point and sometimes, asking an important question, and giving an incomplete (but non-trivial) answer is more interesting and impactful in the long term. We realize this is highly subjective, and hope that you share our taste in research and our work could as reviewer woi7 says, “with the rigor and simplicity … can act as a foundation to build OOV research”.
>
> > It would be helpful to include more intuitive discussion throughout the paper providing more analysis on the sections. For example, more discussion on the assumptions of the settings/theorems would be helpful, and it's not clear exactly under what assumptions the proposed predictor is appropriate.
>
> We thank the reviewer for the suggestion and have included
> * a bullet list of assumptions on theorems and identifiability of the proposed predictors (Appendix E.2)
> * a discussion on violations to each of the assumptions from causal graph, to function class and to noise distributions (Appendix E.2.1).
> * On experimental analysis side, in addition to experiments on real world dataset (Appendix D.4), we conducted additional experiments on the robustness of our approaches when noise level changes (Appendix D.5), and when the distribution of noise is heavy tailed (Appendix D.6)
>
> Overall, we thank you for taking time, which we carefully take into account your feedback and included both experimental results and theoretical discussions in the updated version. We hope that our response has addressed all your questions and concerns. We kindly ask you to let us know if you have any remaining criticism, and - if we have answered your questions - to consider reevaluating your score.

---

### Official Review · Reviewer_s9Ga · 2023-10-31

**Soundness:** 3 good
**Presentation:** 4 excellent
**Contribution:** 2 fair
**Rating:** 8
**Confidence:** 4

**Summary:**

This work investigates out-of-variable (OOV) generalization, which is a sub-problem to OOD generalization, and refers to scenarios where an agent needs to generalize to environments containing variables that were never jointly observed before. The paper shows that if the source and target environments contain some overlapping variables (and under certain conditions), information from the predictor in the source environment can improve predictions in the target environment. More specifically,  the moments of the residual distribution from the optimal classifier in the source environment can be used to calculate the generating function with respect to the unobserved variable in the target domain.

Based on this observation, the paper proposes a practical algorithm for OOV prediction, evaluates its performance, and compares it against the marginal predictor and imputed predictor, as well as an Oracle predictor.

**Strengths:**

The paper proposes a new and important problem-setting - OOV generalization, which can occur in real-world situations, on its own or alongside OOD aspects. The work also provides an extensive study of the identification problems of various variants of OOV scenarios, including theoretical proofs and examples.

In addition, the paper proposes a practical algorithm to solve several OOV scenarios that achieves non-trivial OOV transfer on synthetic data.

The ideas presented in the paper are novel and the conclusion that information from source domains can be used for prediction in the target domain in this setting is important, and can potentially have a broad impact on future research in the field.

**Weaknesses:**

The main limitation of the paper is that the proposed approach was tested on only synthetic data, and was not validated using more challenging datasets.

In addition, the extension of OOV in multi-environments is mentioned mainly in the appendix and the algorithm was not tested empirically for that extension.

**Questions:**

I would like to ask the following questions:

1. For future work, is there a more complicated/realistic dataset to validate the algorithm?
2. Is it possible to compare the algorithm to state-of-the-art marginal or causal methods such as Mejia et al. (2021) or Janzing (2018)? To validate if Vapnik’s principle holds and whether the proposed approach indeed improves results due to solving a less general problem.
3. Theorem 3 connects all moments of the residual distribution to the partial derivatives with respect to the unique variable of the target environment. If additional moments were to be calculated as part of the proposed algorithm, would it improve results (for the general function case)?
4. In general, since the paper's main claim is that in the real world, it is likely to encounter both aspects of OOD and OOV - How simple is it to combine state-of-the-art  OOD methods with the proposed approach? I cannot imagine at the moment a straightforward way to do that.

---

> ### Author Response · Authors · 2023-11-14
>
> > For future work, is there a more complicated/realistic dataset to validate the algorithm?
>
> We thank the reviewer for the suggestion and included results on real-world dataset in Table 2 (Appendix D.4).
>
> > Is it possible to compare the algorithm to state-of-the-art marginal or causal methods such as Mejia et al. (2021) or Janzing (2018)? To validate if Vapnik’s principle holds and whether the proposed approach indeed improves results due to solving a less general problem.
>
> We thank the reviewer for the suggestion, though we consider Janzing (2018)’s work is more on the theoretical fronts and Mejia et al. (2021)’s elegant analysis considers a merging dataset problem which does not directly fit to our particular setting.
>
> > Theorem 3 connects all moments of the residual distribution to the partial derivatives with respect to the unique variable of the target environment. If additional moments were to be calculated as part of the proposed algorithm, would it improve results (for the general function case)?
>
> Thank you for the question, yes incorporating higher-order moments would improve the result as it allows us to estimate general function beyond first-order Taylor approximations. The higher moments provide a space of system of equations that can help to identify the first order partial derivatives and higher order partial derivatives, which assist in estimating a more complex function class.
>
> > In general, since the paper's main claim is that in the real world, it is likely to encounter both aspects of OOD and OOV - How simple is it to combine state-of-the-art OOD methods with the proposed approach? I cannot imagine at the moment a straightforward way to do that.
>
> The reviewer asks re insights on combining both OOD and OOV approaches: we think sparse mechanism shift (SMS) might be a useful notion to characterize distributions shifts as SMS and our current approach both uses causal framework to assist analysis. It assumes the distribution shifts are caused by sparse causal mechanism shifts. We think depending on different variables to which the distribution shift happens would call for a different approach. For example, say, a functional relationship on $Y$ is $\alpha X_1 + X_2 + X_2X_3$. Changing $\alpha$ between the source and target environment does not affect our proposed approach on OOV generalization. However, if $\alpha$ is placed on $X_2X_3$ term, a more sophisticated method need to be developed. We do not have broader results yet, and we feel that the paper is already useful to the community as a starting point to tackle real-world problems that exhibit both OOD and OOV.

---

> ### Comment · Reviewer_s9Ga · 2023-11-22
>
> I thank the authors for answering my questions and appreciate their effort in providing additional experiments.
> My main concern regarding the applicability of the proposed OOV setting to real-world scenarios was partially addressed by the additional experiment presented in Table 2 of the rebuttal. The experiment clearly validates the superiority of the proposed algorithms. However, a more convincing argument would be to validate the proposed approach on more than one real-world dataset (and perhaps a more challenging one).
> Furthermore, since most real-world scenarios are susceptible to aspects of OOV and OOD, I think that the paper would benefit from validating the feasibility of combining the proposed approach with a leading OOD algorithm.
>
> With the above in mind, the paper is well written and I recognize its potential value for the community. For these reasons, I’m updating my score.

---

### Official Review · Reviewer_woi7 · 2023-10-31

**Soundness:** 3 good
**Presentation:** 3 good
**Contribution:** 2 fair
**Rating:** 6
**Confidence:** 3

**Summary:**

This paper describes the out-of-variable OOV problem, which in its simplest form, aims to learn a predictor Y = f_t(X2, X3) given an OOV predictor Y = f_s(X1, X2) and a dataset (X2, X3), but without any instance of (X2, X3, Y). The authors describe the setting in which this is possible and develops an algorithm. The key observation is that the third moment of the residue Y - f_s(X1,X2) contains information about X3 that is least polluted by the noise.

**Strengths:**

- The key observation/discovery is clever, and the algorithm is straight-forward to use.
- The writing is clear, clean, and well-referenced. The examples also made things concrete and easy to follow.
- The rigor and simplicity of the work can act as a foundation to build OOV research.

**Weaknesses:**

- The main weakness is the applicability of the method. The authors only showed results for proof-of-concept, not for real-world usage.
- It is unclear how one could identify whether the assumptions are satisfied given a dataset.
- It is unclear how bad the predictor would be if the assumptions are not satisfied.
- It is not yet clear what realistic problem can be well modeled by OOV generalization.

**Questions:**

Intro:
- It seems OOV fits very well the frame of missing-not-at-random and covariate-dependent missingness. Could the authors comment on that?

Section 2:
- Theorem 2 is slightly confusing for me at first glance because I thought PA_Y by definition includes all parents of Y (so x1,x2, x3 in the example) and not just those in the target environment (x2, x3). It may be helpful to clarify.

Section 3:
As I am trying to get a sense of the restriction and applicability of the approach, I was wondering the following questions:
- How does the method fair with the oracle as the magnitude of the noise increases?
- What if the noise is not gaussian but more heavy tailed?
- Does the performance degrade or improve with increasing number of variables?
- I assume Theorem 3 does not apply to discrete variables because of the violation of differentiability; is that right?

Section 4:
- Can include missing-not-at-random imputation and covariate-missing imputation as two more baseline models (a search in Google scholar using the two key phrases yields some methods).
- It would be really interesting if the authors could find some real-world datasets, create source and target environments by sub-setting the columns, and see how the method performs.
- Figure 3: I don’t quite understand the figure. It would be helpful to define OOV loss, be explicit about the number of samples on the y-axis being (x2,x3,y) or (x1,x2,y) or something else. I also don’t understand why relative loss is zero means the method is on par with the oracle predictor. Why not just show how the fine-tuning error compares with oracle training, which seems easier to interpret? Anyway, I am overall a bit confused about the figure, so my questions may not make sense.

---

> ### Author Response · Authors · 2023-11-14
>
> > It seems OOV fits very well the frame of missing-not-at-random and covariate-dependent missingness. Could the authors comment on that?
>
> We thank the reviewer for the question. Missing-not-at-random and covariate-dependent missingness refer to a scenario where whether a variable is observed or not contains information about other covariates, or about certain other properties of the data point. One can thus hope to exploit this assumption to recover more information about the dataset. We agree that such problems are related to OOV, though think OOV addresses a larger set of problems. For example, the setting of our paper (Fig. 1a) studies when covariates are independent. Such assumption is even more challenging as it provides less information on the remaining dataset. We surprisingly find that even in such challenging scenario, one can enable OOV generalization for certain cases.
> > Theorem 2 is slightly confusing for me at first glance because I thought PA_Y by definition includes all parents of Y (so x1,x2, x3 in the example) and not just those in the target environment (x2, x3). It may be helpful to clarify.
>
> Thank you for your feedback on improving the paper. We have incorporated an explanation on the difference of Theorem 2 setting with the original setup in the paragraph between section 3.3 and section 3.3.1 in the updated version.
>
> > How does the method fair with the oracle as the magnitude of the noise increases?
>
> Thank you for the question, we performed an additional systematic analysis similar in Table 1 with increase in noise standard deviation from 0.01 to 1 in an interval of 0.2. We averaged results over 5 repeated runs and observe our method are robust to increasing noise level and consistently outperforms remaining benchmarks despite noise increases. Table 3 (Appendix D.5) in the updated pdf shows the results.
>
> > What if the noise is not gaussian but more heavy tailed?
>
> Thank you for your question, we included an additional systemic noise when noise is heavy tailed. Specifically, when noise follows a log-normal distribution with mean 0 and sigma 0.5. We repeated the experiment over 5 runs and averaged over a hyperparameter sweep. Table 4 (Appendix D.6) shows the results. We observe as expected by our theorems (Theorem 3), the skew of the noise pollutes the skew estimation of the residual distribution, which deteriorates performance.
>
> > Does the performance degrade or improve with increasing number of variables?
>
> Thank you for your question. When the number of variables increases in the source environment, we expect the performance remains the same as the base network $f_s$’s accuracy is indifferent to the number of variables given sufficient sample size; when the number of variables increases in the space of missing variables, we would expect our method would still be able to transfer under OOV setting. This is due to one can leverage multiple moments from the residual distribution to create a system of equations and solve multiple unknowns. We provide an additional theoretical analysis on a toy example facing two unobserved variables from the source environment in Appendix C.6.
>
> > I assume Theorem 3 does not apply to discrete variables because of the violation of differentiability; is that right?
>
> Yes that is correct.
>
> > Can include missing-not-at-random imputation and covariate-missing imputation as two more baseline models (a search in Google scholar using the two key phrases yields some methods).
>
> Thank for the suggestion, though (re discussion above) missing not at random and covariate dependent imputation do not apply in our setting as our assumptions are different. We assume our covariates to be independent whereas the proposed methods assume covariates are dependent.
>
> > Figure 3: I don’t quite understand the figure. It would be helpful to define OOV loss, be explicit about the number of samples on the y-axis being (x2,x3,y) or (x1,x2,y) or something else. I also don’t understand why relative loss is zero means the method is on par with the oracle predictor. Why not just show how the fine-tuning error compares with oracle training, which seems easier to interpret? Anyway, I am overall a bit confused about the figure, so my questions may not make sense.
>
> We apologize for the confusion, the number of samples on the x-axis refer to (x2, x3, y), and the relative loss is calculated as $log(loss_{pred}/loss_{oracle})$, the log ratio of predictor’s loss divided by the oracle’s loss. If the predictor achieves the same loss as the oracle loss then the ratio would be 1 and $log(1) = 0$. The purpose of the experiment is to show what is the number of samples needed in the target environment, at which we would prefer the MomentLearn over the joint predictor. Figure 3 shows that up to ~100 samples in the target environment, MomentLearn outperforms the joint predictor.

---

> > ### Author Response · Authors · 2023-11-14
> >
> > > It is not yet clear what realistic problem can be well modeled by OOV generalization.
> >
> > We think OOV generalization is a general ability present in both human and animal’s navigation towards Nature. For example, AI in medicine is a relevant area where we often face strong limitations in guaranteeing dataset consistency. To begin with, patients have unique circumstances, and some diseases/symptoms are rare. More generally, the world is full of medical datasets with varying variable sets - diagnostic measurements greatly vary across patients (serum, various forms of imaging, genomics, proteomics, immunoassays, etc.). A good human doctor can generalize across patients even if the measured variables are not identical.
> >
> > To be more concrete, we present a realistic problem in Section 3.1: consider two medical labs collecting different sets of variables. Lab A collects $X_{1}= $ lifestyle factors and $X_2 = $ blood test; Lab B, in addition to $X_2$, collects $X_{3} = $ genomics. Lab A is hospital-based and can measure diseases $Y$, whereas B is a research lab and hope to utilize existing Lab A’s data. The OOV problem asks: given a model trained to predict $Y$ on Lab A’s data, how should Lab B use this model for its own dataset that differs in the set of input variables?
> >
> >
> > Overall, there has been much talk about the potential of AI in medicine since machines can learn from millions of patients, whereas a doctor may only see a few thousand during their lifetime. In practice, data scientists end up using imputation or simply ignore rarely-measured features. To fully realize the potential of AI in medicine, we believe one needs to solve the OOV problem one way or another.
> >
> > > The main weakness is the applicability of the method. The authors only showed results for proof-of-concept, not for real-world usage.
> >
> > We thank the reviewer for the suggestion and here include results on a real-world dataset in Table 2 (Appendix D.4). We observe our method outperforms the other baselines.
> >
> > Overall, we thank the reviewer for taking the time and pushing us to be more explicit on our method’s robustness and applicability. Following your suggestions, we have included 3 additional experiments: on real-world dataset (Table 2), on robustness of changing noise level (Table 3), on robustness of heavy tailed distribution (Table 4). To further offer clarity in understanding the assumptions of our method, we provide a bullet list of assumptions on our theorems in Appendix E.2 and discusses robustness of our method to violation of each assumptions (Appendix E.2.1)

---

> > > ### Comment · Reviewer_woi7 · 2023-11-18
> > >
> > > Thank the authors for the responses. The 3 extra experiments nicely addressed my main requests. I especially like the experiment on the mtcars dataset. I think this is one step closer to seeing OOV generalization and the proposed method in a real-world application. Having said that, I think the work is still a few steps removed from what I would consider naturally real-world, so I will keep my score as is for the time being.

---

### Official Review · Reviewer_xwQY · 2023-11-08

**Soundness:** 3 good
**Presentation:** 3 good
**Contribution:** 3 good
**Rating:** 8
**Confidence:** 3

**Summary:**

The paper investigates out-of-variable generalization, namely the ability for a predictive model to generalize to target domains in which the agent has never seen the joint variables in the target domain in a single source domain before. Under certain assumptions as well as when these assumptions don't fully hold, the paper shows that the error residual distribution in an environment provides information on the unobserved causal parent variable in this environment, and they use this information to derive an algorithm that performs OOV generalization with source and target domains that have overlapping sets of causal factors.

**Strengths:**

**Originality**
- As far as I know, though the problem the paper addresses is well-known as a significant problem, the paper provides several theoretical results, mathematical derivations, and supports these with simple empirical results that are novel.

**Quality**
- The quality of the paper is high. It addresses a high-value problem in a principled fashion, shows how certain assumptions help obtain certain results and how and in which cases these assumptions can be bypasses while maintain approximately accurate results, and evaluates these cases in terms of loss accuracy as well as sample complexity of its approach versus baseline approaches.
- The paper openly highlights limitations in its work, such as assumptions made for theorems to hold, and proposes prospective future work in multiple avenues. This refreshingly is (1) included at all and (2) doesn't seem like a mere afterthought.

**Clarity**
- The paper is mostly clear in its explanation of motivation, preliminaries, approach, baseline usage, results, and limitations.
- The paper does a great job providing simple, clear real-world examples to elucidate the problem and applications of the various theorems included in multiple cases.

**Significance**
- The significance of the problem the paper addresses is high and the problem is ubiquitous. The approach is promising and can be applied in many real-world settings through Monte-Carlo sampling or similar methods. The paper shows that their approach can perform relatively well in "few"-shot settings though this depends on the number of variables involved and the complexity of the problem.

From what I can tell, this is excellent work that I hope motivates further addressing this *out-of-variable* generalization problem by the research and applied AI community. My only reservation is my limited knowledge on the understanding of and state-of-the-art theoretical and applied approaches addressing this problem.

**Weaknesses:**

- Referring to Figure 1, in the first paragraph in page 3, the claim "it would seem all but impossible...(orange box)" could be better explained.
- In Figure 1, it is unclear whether "With $Y$ not observed in the target domain" is an assumption made or is somehow indicated in the diagram or earlier in the paper. Eventually I realized that it's an assumption made, but the illustration Figure 1a alone isn't enough to show this assumption. This ambiguity may clear for some or compound for some later in Section 3.

**Questions:**

- The abstract states "merely considering differences in data distributions is inadequate for fully capturing differences between learning environments." Doesn't out-of-variable technically fall under out-of-distribution, so shouldn't this be adequate? Perhaps more specificity is needed here.
- The abstract states "Mathematically, out-of-variable generalization requires the efficient re-use of past marginal information..." Why does it require efficient re-use? Could it work with "non-efficient" or inefficient re-use?
- On page 2, should "modal" be "model?"
- On page 6, do you mean "parentheses" instead of "brackets" between Eq (9) and Eq (10)?
- Why is the joint predictor considered an oracle predictor if MomentLearn outperforms it?
- Could you explain why MomentLearn is reliably more sample efficient than the oracle predictor for "few"-shot prediction?

**Details Of Ethics Concerns:**

I have no ethics concerns.

---

> ### Author Response · Authors · 2023-11-14
>
> We thank the reviewer for the suggestions to help us clarify Figure 1. We have incorporated the proposed changes in the updated version, by explaining the claim further and changing the border of the orange box.  We also thank the reviewer for their detailed observations on typos and have corrected them in the updated version.
>
> > The abstract states "merely considering differences in data distributions is inadequate for fully capturing differences between learning environments." Doesn't out-of-variable technically fall under out-of-distribution, so shouldn't this be adequate? Perhaps more specificity is needed here.
>
> Here we refer to settings exhibit in-distribution if the environments share the same data-generating process. For example, in Fig 1a, though the environments share the same data generating process (in-distribution) but observed different sets of variables (out-of-variable). To give an example on OOD but not OOV, consider vaccine data from different countries: they have the same variables [Age, is_vaccined], but coming from different distributions due to country differences. Thank you for your question and we will make this point clearer in the next version.
>
> > Why is the joint predictor considered an oracle predictor if MomentLearn outperforms it?
>
> We apologize for the confusion. In Table 1, we refer to the joint predictor as oracle predictor in cases where there are enough data samples in the target environment such that regressing on the joint variable will lead to a near optimal predictor. In Figure 3, we showed that when the sample size is small (~100 samples) in the target environment, joint predictor can lead to estimation error. MomentLearn predictor thus is able to outperform it.
>
> > Could you explain why MomentLearn is reliably more sample efficient than the oracle predictor for "few"-shot prediction?
>
> MomentLearn is more sample efficient than the joint predictor in “few”-shot prediction because with small sample size, the joint predictor would lead to estimation error, whereas MomentLearn leverages the information observed in the source environment (due to large sample size in source) are able to mitigate the problem of small sample size in the target environment.

---

> > ### Comment · Reviewer_xwQY · 2023-11-23
> >
> > Thanks Authors for addressing the Weaknesses and Questions I presented.
> >
> > I understand that OOD $\nRightarrow$ OOV. My claim is that OOV $\nRightarrow$ OOD (in the interpretation of OOD that nearly everyone defaults to), but it seems like you are implying that you are treating in or out-of-distribution as a environmental property rather than a task, or what is seen by the task, property. Please correct me if I am incorrect and if you're able to. Regardless, it isn't a big deal, as I'm pretty sure I understand your explanation. However, I do think it would be useful to make this more clear, as some readers may have qualms about this.
> >
> > Thanks for explaining your reasoning about why and in which cases you refer to the joint predictor as an oracle predictor. I do see that in the caption for Figure 1(c) the paper states "an oracle solution trained as if we have sufficient data..." I would recommend making this explicit also in the main text when "oracle" is first mentioned in the main body in the first paragraph of 4 Experiments. I see that it says "...large datasets," but you may want to make it at least as clear as you do in the caption and a reminder that the oracle doesn't hold when there isn't enough data or all the variables aren't observed when you introduce Figure 3, if you haven't done something similar already.
> >
> > Thank you for your insight into why MomentLearn is reliably more sample efficient than the oracle predictor for "few"-shot prediction?
> >
> > I will retain my ratings as they are for now. Thank you, Authors! Nice work.

---

### Author Response · Authors · 2023-11-14
**General Response**

We thank all the reviewers for their time and effort in providing valuable feedback. We are glad to see that the reviewers acknowledge our work as studying a “high-value problem in a principled fashion”, with “clear”, “well-referenced” writing and “simple, clear real-world examples”.

We respond to all reviewer’s points in detail but wanted to highlight:
* We followed reviewer woi7 and reviewer s9Ga's suggestions and validated our algorithm on real-world dataset with results in Table 2 (see below). We observe our method outperforms the other benchmarks.
* We followed reviewer woi7’s suggestion and conducted two additional experiments: with changing standard deviation of noise in Table 3 (see below) and with heavy tailed noise distribution in Table 4 (see below). We observe our method is robust with changing noise level but performance decreases with heavy tailed noise distribution. This is expected by our theoretical results (Theorem 3) where the skew of noise pollutes the useful signal.
* We followed reviewer qFZD’s suggestion and provide an additional bullet list of assumptions (Appendix E.2) for our theorems and identifiability result, and included a discussion on violation of each assumption in Appendix E.2.1.

---

> ### Author Response · Authors · 2023-11-15
> **Table 2: real world experiment (details in Appendix D.4)**
>
> Our method's (“MomentLearn”) OOV prediction performance in the target environment, compared to the "Marginal" baseline and the predictor that imputes missing variable with its mean ("Mean Imputed"). Shown are mean and standard deviations of the MSE loss between the predicted and observed target values on augmented 'Mtcars' dataset.
>
> |           | Mtcars          |
> |-------------|-----------------|
> | MomentLearn | 1.09 $\pm$ 0.08 |
> | MeanImputed | 1.48 $\pm$ 0.06 |
> | Marginal    | 1.46 $\pm$ 0.03 |

---

> > ### Author Response · Authors · 2023-11-15
> > **Table 3: Experiment on robustness with changing noise level (details in Appendix D.5)**
> >
> > Under noise sampled from Gaussian distribution with varying standard deviation σ, our method’s (“MomentLearn”) OOV prediction performance in the target environment, compared to the “Marginal” baseline, the predictor that imputes missing variable with its mean (“Mean Imputed”) and the solution that has access to the full joint observations on the target domain (“Oracle”). Shown are mean and standard deviations of the MSE loss between the predicted and observed target values. We observe MomentLearn outperforms the other baselines for almost all cases.
> >
> > | $\sum_i \alpha_i X_i$              | $\sigma=0.01$   | $\sigma=0.2$    | $\sigma=0.4$    | $\sigma=0.6$    | $\sigma=0.8$    | $\sigma=1.0$    |
> > |------------------------------------|-----------------|-----------------|-----------------|-----------------|-----------------|-----------------|
> > | Oracle                             | 0.37 $\pm$ 0.18 | 0.23 $\pm$ 0.15 | 0.28 $\pm$ 0.13 | 0.83 $\pm$ 0.38 | 0.80 $\pm$ 0.14 | 1.05 $\pm$ 0.14 |
> > | MomentLearn                        | 0.36 $\pm$ 0.17 | 0.25 $\pm$ 0.16 | 0.32 $\pm$ 0.15 | 0.90 $\pm$ 0.36 | 0.86 $\pm$ 0.12 | 1.13 $\pm$ 0.21 |
> > | MeanImputed                        | 0.68 $\pm$ 0.52 | 0.38 $\pm$ 0.22 | 0.34 $\pm$ 0.18 | 0.94 $\pm$ 0.52 | 0.96 $\pm$ 0.16 | 1.49 $\pm$ 0.68 |
> > | Marginal                           | 0.76 $\pm$ 0.64 | 0.42 $\pm$ 0.25 | 0.36 $\pm$ 0.21 | 0.88 $\pm$ 0.45 | 1.01 $\pm$ 0.20 | 1.61 $\pm$ 0.85 |
> > | $+\sum_{i<j} \beta_{ij} X_i X_j$ | $\sigma=0.01$   | $\sigma=0.2$    | $\sigma=0.4$    | $\sigma=0.6$    | $\sigma=0.8$    | $\sigma=1.0$    |
> > | Oracle                             | 0.26 $\pm$ 0.22 | 0.33 $\pm$ 0.23 | 0.53 $\pm$ 0.28 | 0.52 $\pm$ 0.08 | 0.85 $\pm$ 0.32 | 1.10 $\pm$ 0.07 |
> > | MomentLearn                        | 0.33 $\pm$ 0.21 | 0.79 $\pm$ 0.89 | 0.55 $\pm$ 0.29 | 0.73 $\pm$ 0.43 | 1.05 $\pm$ 0.37 | 1.38 $\pm$ 0.33 |
> > | MeanImputed                        | 0.45 $\pm$ 0.31 | 0.82 $\pm$ 0.50 | 1.21 $\pm$ 1.03 | 1.02 $\pm$ 0.84 | 1.10 $\pm$ 0.39 | 1.39 $\pm$ 0.23 |
> > | Marginal                           | 0.54 $\pm$ 0.43 | 0.87 $\pm$ 0.46 | 1.42 $\pm$ 1.10 | 1.14 $\pm$ 1.12 | 1.16 $\pm$ 0.42 | 1.46 $\pm$ 0.19 |
> > | $+\sum_i \gamma_i X_i^2 $     | $\sigma=0.01$   | $\sigma=0.2$    | $\sigma=0.4$    | $\sigma=0.6$    | $\sigma=0.8$    | $\sigma=1.0$    |
> > | Oracle                             | 1.05 $\pm$ 1.33 | 0.44 $\pm$ 0.43 | 0.51 $\pm$ 0.27 | 0.68 $\pm$ 0.13 | 1.31 $\pm$ 0.78 | 1.74 $\pm$ 0.41 |
> > | MomentLearn                        | 1.39 $\pm$ 1.47 | 0.70 $\pm$ 0.61 | 0.67 $\pm$ 0.33 | 1.04 $\pm$ 0.32 | 1.68 $\pm$ 1.15 | 1.84 $\pm$ 0.47 |
> > | MeanImputed                        | 1.41 $\pm$ 1.22 | 0.80 $\pm$ 0.53 | 0.99 $\pm$ 0.35 | 1.42 $\pm$ 0.70 | 1.44 $\pm$ 0.80 | 1.99 $\pm$ 0.24 |
> > | Marginal                           | 1.62 $\pm$ 1.19 | 0.86 $\pm$ 0.57 | 1.01 $\pm$ 0.37 | 1.39 $\pm$ 0.53 | 1.57 $\pm$ 0.84 | 2.08 $\pm$ 0.27 |   |

---

> > > ### Author Response · Authors · 2023-11-15
> > > **Table 4: Experiment on robustness with heavy tailed noise (details in Appendix D.6)**
> > >
> > > Under heavy tailed noise sampled from log-normal distribution with $\mu = 0$ and $\sigma = 0.5$, our method’s (“MomentLearn”) OOV prediction performance in the target environment, compared to the “Marginal” baseline, the predictor that imputes missing variable with its mean (“Mean Imputed”) and the solution that has access to the full joint observations on the target domain (“Oracle”). Shown are mean and standard deviations of the MSE loss between the predicted and observed target values. $\mathcal{GP}_i(·)$ denotes a function sampled from a Gaussian Process with zero mean and Gaussian kernel. We observe our method shows a decrease in performance as expected by our theoretical result (Theorem 3).
> > >
> > > |           | $\sum_i \alpha_i X_i$                     | $+\sum_{i<j} \beta_{ij} X_i X_j$       | $+\sum_i \gamma_i X_i^2 $                 |
> > > |-------------|---------------------------------------------|------------------------------------------|---------------------------------------------|
> > > | Oracle      | 0.69 $\pm$ 0.14                             | 0.63 $\pm$ 0.43                          | 0.93 $\pm$ 0.33                             |
> > > | MomentLearn | 0.91 $\pm$ 0.34                             | 0.97 $\pm$ 0.50                          | 1.20 $\pm$ 0.30                             |
> > > | MeanImputed | 0.89 $\pm$ 0.31                             | 1.00 $\pm$ 0.53                          | 1.76 $\pm$ 0.86                             |
> > > | Marginal    | 0.93 $\pm$ 0.23                             | 0.90 $\pm$ 0.45                          | 2.11 $\pm$ 1.30                             |
> > > |          | $\mathcal{GP}_1(X_1, X_2) + \alpha_3 X_3$ | $+ \mathcal{GP}_2(X_1, X_2) \cdot X_3$ | $+ \mathcal{GP}_3(X_1, X_2) \cdot X_3^2 $ |
> > > | Oracle      | 0.39 $\pm$ 0.03                             | 0.44 $\pm$ 0.08                          | 0.54 $\pm$ 0.16                             |
> > > | MomentLearn | 0.82 $\pm$ 0.35                             | 0.72 $\pm$ 0.21                          | 1.40 $\pm$ 0.66                             |
> > > | MeanImputed | 0.85$\pm$ 0.42                              | 0.78 $\pm$ 0.41                          | 1.31 $\pm$ 0.95                             |
> > > | Marginal    | 0.65 $\pm$ 0.15                             | 0.79 $\pm$ 0.39                          | 1.81 $\pm$ 1.22                             |

---

### Comment · Area_Chair_WQTQ · 2023-11-22

Dear reviewers,

This a reminder that deadline of author/reviewer discussion is AOE Nov 22nd (today). Please engage in the discussion, check if your concerns are addressed, and make potential adjustments to the rating and reviews.

Thank you!
AC

---

### Meta-Review · Area_Chair_WQTQ · 2023-12-05

**Metareview:**

This paper studies the out-of-variable generalization problem where the source and target domains don't necessarily share the same set of variables. A key observation in this work is that the moments of the residual distribution of the optimal predictor in the source domain carry information about the missing variable in the target domain. This work also proposes a new method for OOV prediction based on this key observation. Overall, reviewers share the concern that the experiments are insufficient compared to typical machine learning papers, and it is unclear what kind of real world problems the proposed method can solve. However, the solution is very solid and inspiring for the community to build upon and brings interesting perspectives to this important OOV generalization problem. The exposition is clear and the theoretical results are intriguing.

During the discussion phase, the reviewers reached consensus to accept this paper and I concur. Please incorporate all reviewer feedback in the revision and add the new experiment results to the main paper.

**Justification For Why Not Higher Score:**

Overall, reviewers share the concern that the experiments are insufficient compared to typical machine learning papers, and it is unclear what kind of real world problems the proposed method can solve.

**Justification For Why Not Lower Score:**

The solution is very solid and inspiring for the community to build upon and brings interesting perspectives to this important OOV generalization problem. The exposition is clear and the theoretical results are intriguing.

---

### Decision · Program_Chairs · 2024-01-16

Accept (poster)